# Generalizing Causal Effects from Randomized Controlled Trials to Target Populations across Diverse Environments

**Baohong Li** [1] **Yingrong Wang** [1] **Anpeng Wu** [1] **Ming Ma** [2] **Ruoxuan Xiong** [3] **Kun Kuang** [1]

## Abstract

Generalizing causal effects from Randomized Controlled Trials (RCTs) to target populations across diverse environments is of significant practical importance, as RCTs are often costly and logistically complex to conduct. A key challenge is environmental shift, defined as changes in the distribution and availability of covariates between source and target environments. A common approach addressing this challenge is to identify a separating set–covariates that govern both treatment effect heterogeneity and environmental differences–and combine RCT samples with target populations matched on this set. However, this approach assumes that the separating set is fully observed and shared across datasets, an assumption often violated in practice. We propose a novel Two-Stage Doubly Robust (2SDR) method that relaxes this assumption by allowing the separating set to be observed in only one of the two datasets. 2SDR leverages shadow variables to impute missing components of the separating set and generalize treatment effects across environments in a two-stage procedure. We show the identification of causal effects in target environments under 2SDR and demonstrate its effectiveness through extensive experiments on both synthetic and real-world datasets.

## 1. Introduction

Estimating treatment effects is crucial for informing whether to apply a new policy or treatment in a given population (the *target population*) (Hernán & Robins, 2020; Luo et al., 2024). Randomized Controlled Trials (RCT) are generally

[1]College of Computer Science and Technology, Zhejiang University, Hangzhou, China [2]Kuaishou, Beijing, China [3]Department of Quantitative Theory & Methods, Emory University, Atlanta, USA. Correspondence to: Kun Kuang <kunkuang@zju.edu.cn>.

*Proceedings of the 42nd International Conference on Machine Learning*, Vancouver, Canada. PMLR 267, 2025. Copyright 2025 by the author(s).

considered the gold standard for estimating causal effects of treatments (Imbens & Rubin, 2015; Hariton & Locascio, 2018). However, due to the long duration and high cost of RCTs, it is often desired to generalize the treatment effect estimates from existing RCTs to target populations across different environments.

A major challenge in generalizing RCT findings across environments is environmental shifts, defined as changes in the distribution and availability of covariates between the RCT sample and the target population (Kuang et al., 2018; 2020). Such shifts cause RCT participants to no longer represent the target population, thereby limiting the external validity and generalizability of causal effect estimates from the RCT (Nguyen et al., 2018; Colnet et al., 2024).

A common solution to addressing the challenge of environmental shifts is to combine RCT data with observational data from the target population, using a *separating set* for adjustment. Compared to collecting new RCT data, obtaining observational data from the target population is typically more cost-effective and feasible. The separating set consists of covariates that simultaneously affect both treatment effect heterogeneity and environmental shifts. Prior work has shown that the Target population Average Treatment Effect (TATE) is identifiable with the separating set (Cole & Stuart, 2010; Tipton, 2013; Kern et al., 2016; Egami & Hartman, 2021; Pearl & Bareinboim, 2022), and many methods have been proposed to leverage the separating set for generalizing treatment effects from RCTs to target populations (Stuart et al., 2011; Hartman et al., 2015; Lesko et al., 2017; Dahabreh et al., 2019; Lee et al., 2023).

The limitation of this common approach is its reliance on the assumption that the covariates shared across both datasets fully contain the separating set–an assumption that is often violated in real-world settings with environmental shifts. When data from the RCT and the target population are collected in different environments, it is challenging to ensure complete alignment in the covariates measured, leading to a *missing covariates* problem: certain covariates present in one dataset may be entirely absent in the other. If these missing covariates include variables from the separating set, existing methods may fail to generalize treatment effects accurately, as shown in prior studies (Nguyen et al., 2017;

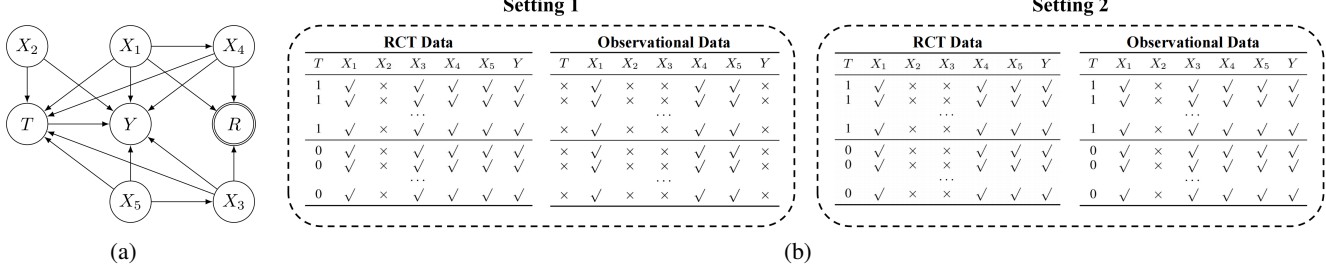

*Figure 1.* (a) The causal graph for the problem setup. For clarity, we only present the causal graph for the observational data, while for the RCT data, all edges pointing to $T$ are removed because the treatment is randomly assigned. Suppose that the heterogeneous treatment effect of $T$ on $Y$ is $Y(1) - Y(0) = \tau(X_1, X_2, X_3)$, where $\tau(X_1, X_2, X_3)$ is a function of $X_1$, $X_2$, and $X_3$. $\{X_1, X_2, X_3\}$ is the set of covariates affecting treatment effect heterogeneity. $\{X_1, X_3, X_4\}$ is the set of covariates affecting environmental shifts (pointing to $R$). $X_2$ is a covariate affecting neither treatment effect heterogeneity nor environmental shifts. Therefore, the separating set, which contains all variables that simultaneously affect treatment effect heterogeneity and environmental shifts, is $\mathbf{W} = \{X_1, X_3\}$. (b) Example data for the problem setup, where $\mathbf{X}^{c} = \{X_1, X_4, X_5\}$ is the set of common covariates observed in both datasets, $\mathbf{X}^{m} = \{X_3\}$ is the set of covariates missing in one of the datasets, $\mathbf{Z} = \{X_5\}$ is the shadow variable set of $\mathbf{X}^{m}$, which are correlated with the $\mathbf{X}^{m}$ but do not directly influence environmental shifts, and $\widetilde{\mathbf{X}^{c}} = \{X_1, X_4\}$ denotes $\mathbf{X}^{c} \setminus \mathbf{Z}$.

Andrews & Oster, 2019; Colnet et al., 2022; Dahabreh et al., 2023; Huang, 2024) and supported by our experiments.

We propose a solution to relax the assumption required by the common approach. Specifically, we only require variables from the separating set to be observed in only one of the two datasets–either the RCT data or the observational data from the target population, as illustrated in Figure 1.

In Setting 1, only the covariates from the target population are available or used. The RCT data include all covariates from the separating set, but some of these covariates are missing in the target population data. For example, in evaluating the treatment effect of a new drug on AIDS, RCTs typically collect various clinical variables rarely measured in routine check-ups, such as the CD4 count, which is an important treatment effect modifier and a variable in the separating set. Since the new drug has not yet been publicly used, the observational data from the target population lack treatment and outcome information and only include basic demographic covariates, such as gender and race, and omit those rarely measured variables available in the RCT data (Hammer et al., 1997; Prejean et al., 2008; Hall et al., 2008).

In Setting 2, the observational data from the target population contain treatment and outcome information, along with all covariates from the separating set. However, some of these covariates are missing in the RCT data. For example, when RCTs are conducted at small sites, limitations such as long tracking periods, budget constraints, and lack of equipment may result in insufficient covariate collection. In contrast, when observational data from the target population are collected at larger sites, treatment and outcome information, along with more comprehensive covariates, can be

obtained through large-scale surveys or interviews (Collaboration, 2009; Resche-Rigon et al., 2013; Jolani et al., 2015; Huang et al., 2023; Huang, 2024).

Given this relaxed assumption, our proposed method, Two-Stage Doubly Robust (2SDR), leverages *shadow variables* to impute the missing covariates and generalize treatment effects from RCTs to target populations across environments via a two-stage procedure. In the first stage, we introduce a feature selection method to identify shadow variables of the missing covariates, enabling unbiased imputation of these covariates in a doubly robust manner. In the second stage, we apply a doubly robust approach to obtain an unbiased estimate of TATE. We show the identification of causal effects in target environments under this approach and demonstrate its effectiveness through extensive experiments on both synthetic and real-world datasets.

In summary, the contributions of this paper are as follows:

- We propose a novel identification framework for the generalizability of treatment effects from RCTs to target populations across diverse environments, relaxing the assumptions in prior work.
- We develop a novel method for generalizing treatment effects to target populations under environmental shifts, ensuring an unbiased estimate of the TATE.
- We demonstrate the effectiveness of our method through extensive experiments on synthetic data and applications to two real-world datasets.

## 2. Preliminaries

### 2.1. Problem Formulation

Throughout the paper, $T$ denotes the binary treatment variable, where $t_i = 1$ indicates that unit $i$ is assigned to the treatment, and $t_i = 0$ indicates otherwise. $Y$ is the outcome variable, and $Y(t)$ denotes the potential outcome under $T = t$ (Rubin, 1974). $\mathbf{X} = \{\mathbf{X}^c, \mathbf{X}^m\}$ is the pretreatment covariates, where $\mathbf{X}^c$ denotes the *common covariates* present in both datasets, and $\mathbf{X}^m$ denotes the *covariates missing* in one of the two datasets. $S$ is a binary variable indicating in which dataset $\mathbf{X}^m$ is observable, where $s_i = 1$ indicates that $\mathbf{X}^m$ is observed in the RCT data, and $s_i = 0$ indicates otherwise. $R$ is a binary variable indicating which dataset a unit belongs to, where $r_i = 1$ indicates that unit $i$ belongs to the RCT data, and $r_i = 0$ indicates otherwise.

For simplicity of exposition, we introduce our solution in the context of generalizing an RCT to the target population of one environment. However, our solution can be extended to multiple target populations in various environments. In Setting 1, the RCT data consists of $n_\mathcal{R}$ independent random tuples $\{\mathbf{x}_i, y_i, t_i, s_i = 1, r_i = 1\}_{i=1}^{n_\mathcal{R}}$ from all participants in the RCT, denoted by $\mathcal{R}$, while the observational data from the target population consists of $n_\mathcal{O}$ independent random tuples $\{\mathbf{x}_i, s_i = 0, r_i = 0\}_{i=1}^{n_\mathcal{O}}$ randomly drawn from the target population, denoted by $\mathcal{O}$. In Setting 2, the RCT data consists of $\{\mathbf{x}_i, y_i, t_i, s_i = 0, r_i = 1\}_{i=1}^{n_\mathcal{R}}$, while the the observational data from the target population consists of $\{\mathbf{x}_i, y_i, t_i, s_i = 1, r_i = 0\}_{i=1}^{n_\mathcal{O}}$. The total number of samples in both datasets is denoted by $n = n_\mathcal{R} + n_\mathcal{O}$. In this paper, we consider the *non-nested trial design* scenario, where the two datasets are collected separately from different environments with environmental shifts, which is defined as follows (Kuang et al., 2018; 2020; Dahabreh et al., 2020; 2021).

**Definition 2.1. (Environmental Shifts.)** Environmental shifts refer to the shifts in the distribution and quantity of covariates between Dataset $\mathcal{R}$ and Dataset $\mathcal{O}$, i.e., $\mathbb{P}(\mathbf{X} \mid R = 1) \neq \mathbb{P}(\mathbf{X} \mid R = 0)$, and $\mathbf{X}^m$ is unobserved in one of the two datasets.

The specific scenarios of environmental shifts applicable to this paper are discussed in Appendix B.

We are interested in estimating the average treatment effect in the target population. This causal estimand is called the Target population Average Treatment Effect (TATE) and is defined as follows (Tipton, 2013; Kern et al., 2016).

**Definition 2.2. (Target population Average Treatment Effect.)** $\tau = \mathbb{E}[Y(1) - Y(0) \mid R = 0]$.

The TATE cannot be directly estimated using Dataset $\mathcal{O}$, as the values of the treatment are totally unobserved. Fortunately, combining Dataset $\mathcal{O}$ with Dataset $\mathcal{R}$ makes the TATE identifiable under certain assumptions.

### 2.2. TATE Identifiability Conditions

First, to ensure the identifiability of the average treatment effect within Dataset $\mathcal{R}$, we make the following assumptions about $\mathcal{R}$ (Cole & Stuart, 2010; Stuart et al., 2011; Imbens & Rubin, 2015; Colnet et al., 2022; Dahabreh et al., 2023).

**Assumption 2.3. (Treatment randomization within the RCT.)** $Y(1), Y(0) \perp\!\!\!\perp T \mid R = 1$.

**Assumption 2.4. (Positivity of trial treatment assignment.)** $0 < \mathbb{P}(T = 1 \mid \mathbf{X}, R = 1) < 1$.

**Assumption 2.5. (Positivity of trial participation.)** $0 < \mathbb{P}(R = 1 \mid \mathbf{X}) < 1$.

**Assumption 2.6. (Stable Unit Treatment Value Assumption.)** The distribution of the potential outcome of one unit is independent of the treatment assignment of another unit.

Under the above assumptions, the average treatment effect of Dataset $\mathcal{R}$ is identifiable. Next, we formally introduce the conditions for the generalizability of the RCT estimates, i.e., the identifiability conditions of the TATE (Cole & Stuart, 2010; Tipton, 2013; Kern et al., 2016; Egami & Hartman, 2021; Pearl & Bareinboim, 2022).

**Definition 2.7. (Separating Set.)** A separating set is a set that contains all variables that simultaneously affect treatment effect heterogeneity and environmental shifts, i.e., $Y(1) - Y(0) \perp\!\!\!\perp R \mid \mathbf{W}$.

**Assumption 2.8. (Fully observability of the separating set.)** The covariates common to both datasets include all the variables from the separating set, i.e., $\mathbf{W} \subset \mathbf{X}^c$.

**Theorem 2.9.** *(Identification of the TATE.) Under Assumptions 2.3, 2.4, 2.5, 2.6, and 2.8, the TATE is identified as*

$$\tau = \int \left(\mathbb{E}[Y(1) \mid \mathbf{W} = \mathbf{w}] - \mathbb{E}[Y(0) \mid \mathbf{W} = \mathbf{w}]\right)$$
$$dF(\mathbf{W} = \mathbf{w} \mid R = 0),$$

*where $F(\mathbf{W} \mid R = 0)$ is the cumulative distribution function of $\mathbf{W}$ conditional on $R = 0$, and $\mathbb{E}[Y(t) \mid \mathbf{W} = \mathbf{w}_i] = \mathbb{E}[Y \mid \mathbf{W} = \mathbf{w}_i, T = t, R = 1]$.*

Prior work proposes different ways to adjust for $\mathbf{W}$ based on Theorem 2.9 (Hartman et al., 2015; Dahabreh et al., 2019; Lee et al., 2023). However, because of environmental shifts, it is difficult to ensure that all variables from the separating set are observed in both datasets, making Assumption 2.8 violated and existing method ineffective (Nguyen et al., 2017; Colnet et al., 2022; Dahabreh et al., 2023).

## 3. Extended TATE Identifiability Conditions

In this paper, we propose a novel TATE identifiability framework that relaxes the requirement for the separating set to be observable in both datasets to requiring it to be observable in at least one. The key assumption is as follows:

**Assumption 3.1. (Partial observability of the separating set.)** All variables in the separating set are observable in at least one of Dataset $\mathcal{R}$ or Dataset $\mathcal{O}$, i.e., $\mathbf{W} \subset \mathbf{X}$.

To formalize this, we introduce an indicator variable $S$, where $S = 1$ if the separating set is observed and $S = 0$ if it is missing. As illustrated in Figure 1, Assumption 3.1 indicates one of the following two settings:

- **(Setting 1.)** Dataset $\mathcal{O}$ has no treatment and outcome information. The separating set is observed in Dataset $\mathcal{R}$, i.e., in Dataset $\mathcal{R}$, $S = 1$, and in Dataset $\mathcal{O}$, $S = 0$.
- **(Setting 2.)** Dataset $\mathcal{O}$ contains treatment and outcome information. The separating set is observed in Dataset $\mathcal{O}$, i.e., in Dataset $\mathcal{R}$, $S = 0$, and in Dataset $\mathcal{O}$, $S = 1$.

In Setting 1, the relationship between $S$ and $R$ is $S = R$, whereas in Setting 2, the relationship is $S = 1 - R$. Both settings are common in real-world scenarios (Hammer et al., 1997; Prejean et al., 2008; Collaboration, 2009; Resche-Rigon et al., 2013; Jolani et al., 2015), as illustrated by the two examples provided in Section 1.

**Assumption 3.2. (Shadow variable assumption.)** Among the common covariates $\mathbf{X}^c$ shared by datasets $\mathcal{R}$ and $\mathcal{O}$, there exists a set of covariates $\mathbf{Z} \subset \mathbf{X}^c$ that satisfy the following conditions: (1) $\mathbf{Z}$ is conditionally correlated with $\mathbf{X}^m$, i.e., $\mathbf{Z} \not\perp\!\!\!\perp \mathbf{X}^m \mid \widetilde{\mathbf{X}^c}, S = 1$. (2) $\mathbf{Z}$ is conditionally independent of $S$, i.e., $\mathbf{Z} \perp\!\!\!\perp S \mid \mathbf{X}^m, \widetilde{\mathbf{X}^c}$, where $\widetilde{\mathbf{X}^c}$ denotes the set difference $\mathbf{X}^c \setminus \mathbf{Z}$, i.e., $\mathbf{X}^c = \{\widetilde{\mathbf{X}^c}, \mathbf{Z}\}$.

Assumption 3.2 requires that among the common covariates, there exist shadow variables $\mathbf{Z}$ that are correlated with the covariates $\mathbf{X}^m$ missing in one of the datasets but do not directly influence environmental shifts (as indicated by $S$); see, e.g., $\mathbf{Z} = \{X_5\}$ in Figure 1. This assumption is reasonable in many real-world settings, as not all observed variables impact the environmental shifts. For example, in studies of drug effects on AIDS, body weight can serve as a shadow variable $\mathbf{Z}$: it has been shown to be associated with CD4 count (a missing covariate), but does not directly influence RCT participant selection (Womack et al., 2007) (measured by $S$). Moreover, this assumption is *testable* using observed data (d'Haultfoeuille, 2010; Miao et al., 2024; Li et al., 2024b;c).

**Theorem 3.3. *(Extended TATE Identifiability Conditions.)*** *Under Assumptions 2.3, 2.4, 2.5, 2.6, 3.1, and 3.2, if the completeness condition holds[1], the TATE is identified as*

$$\tau = \int \left( \mathbb{E}[Y(1) \mid \mathbf{X} = \mathbf{x}] - \mathbb{E}[Y(0) \mid \mathbf{X} = \mathbf{x}] \right)$$

$$dF(\mathbf{X} = \mathbf{x} \mid R = 0), \tag{1}$$

---

[1] If for all square-integrable functions $h(\mathbf{X}^m, \mathbf{X}^c)$, $\mathbb{E}[h(\mathbf{X}^m, \mathbf{X}^c) \mid \mathbf{X}^m, \mathbf{X}^c, S = 1] = 0$ almost surely if and only if $h(\mathbf{X}^m, \mathbf{X}^c) = 0$ almost surely.

*where $F(\mathbf{X} \mid R = 0)$ denotes the cumulative distribution function of $\mathbf{X}$ conditional on $R = 0$*

*Proof.* To prove the identifiability of Equation (1), we must prove both $F(\mathbf{X} \mid R = 0)$ and $\mathbb{E}[Y(t) \mid \mathbf{X}]$ are identifiable.

**Step 1.** Proof of the identifiability of $F(\mathbf{X} \mid R = 0)$.

**Step 1.1.** Proof in Setting 2 (separating set in $\mathcal{O}$).

In Setting 2, all covariates $\mathbf{X}$ in Dataset $\mathcal{O}$ are observable. Therefore, $F(\mathbf{X} \mid R = 0)$ is directly identifiable.

**Step 1.2.** Proof in Setting 1 (separating set in $\mathcal{R}$).

In Setting 1, however, as the values of $\mathbf{X}^m$ are missing in Dataset $\mathcal{O}$, which may contain variables from the separating set, $\mathbb{P}(\mathbf{X}, R = 0) = \mathbb{P}(\mathbf{X}^m, \mathbf{X}^c, R = 0)$ is not available. Therefore, to prove the identifiability of $F(\mathbf{X} = \mathbf{x} \mid R = 0)$, we must prove $\mathbb{P}(\mathbf{X}^m, \mathbf{X}^c, R = 0)$ is identifiable.

Since $S = R$ in Setting 1, we have $\mathbb{P}(\mathbf{X}^m, \mathbf{X}^c, R = 0) \equiv \mathbb{P}(\mathbf{X}^m, \mathbf{X}^c, S = 0)$. We show that this distribution can be identified using the shadow variables $\mathbf{Z}$ and common covariates $\mathbf{X}^c$. The corresponding lemmas are stated below, with detailed proofs provided in Appendices D and E.

**Lemma 3.4.** *Under Assumptions 2.5 and 3.2, $\mathbb{P}(\mathbf{X}^m \mid \mathbf{X}^c, S = 0)$ is identified as*

$$\mathbb{P}(\mathbf{X}^m \mid \mathbf{X}^c, S = 0) = \frac{\mathrm{OR}(\mathbf{X}^m, \mathbf{X}^c) \cdot \mathbb{P}(\mathbf{X}^m \mid \mathbf{X}^c, S = 1)}{\mathbb{E}[\mathrm{OR}(\mathbf{X}^m, \mathbf{X}^c) \mid \mathbf{X}^c, S = 1]}, \tag{2}$$

*where*

$$\mathrm{OR}(\mathbf{X}^m, \mathbf{X}^c) = \frac{\mathbb{P}(S = 0 \mid \mathbf{X}^m, \mathbf{X}^c)}{\mathbb{P}(S = 1 \mid \mathbf{X}^m, \mathbf{X}^c)} \times \frac{\mathbb{P}(S = 1 \mid \mathbf{X}^m = \mathbf{0}, \mathbf{X}^c)}{\mathbb{P}(S = 0 \mid \mathbf{X}^m = \mathbf{0}, \mathbf{X}^c)}. \tag{3}$$

Here, $\mathrm{OR}(\mathbf{X}^m, \mathbf{X}^c)$ is the odds ratio function, and $\mathbf{0}$ is a reference value, which can be any other value within the value range of $\mathbf{X}^m$. The identification of Equation (2) requires the identification of $\mathbb{P}(\mathbf{X}^m \mid \mathbf{X}^c, S = 1)$ and $\mathrm{OR}(\mathbf{X}^m, \mathbf{X}^c)$. The former is directly identifiable in Dataset $\mathcal{R}$. The latter can be identified based on Lemma 3.5 below, with detailed proofs provided in Appendix E.

**Lemma 3.5.** *Under Assumptions 2.5 and 3.2, if the completeness condition holds, $\mathrm{OR}(\mathbf{X}^m, \mathbf{X}^c)$ is identified as*

$$\mathrm{OR}(\mathbf{X}^m, \mathbf{X}^c) = \widetilde{\mathrm{OR}}(\mathbf{X}^m, \mathbf{X}^c) / \widetilde{\mathrm{OR}}(\mathbf{X}^m = \mathbf{0}, \mathbf{X}^c), \tag{4}$$

*where*

$$\widetilde{\mathrm{OR}}(\mathbf{X}^m, \mathbf{X}^c) = \frac{\mathrm{OR}(\mathbf{X}^m, \mathbf{X}^c)}{\mathbb{E}[\mathrm{OR}(\mathbf{X}^m, \mathbf{X}^c) \mid \widetilde{\mathbf{X}^c}, S = 1]} \tag{5}$$

*is identified by*

$$\mathbb{E}[\widetilde{\mathrm{OR}}(\mathbf{X}^{\mathrm{m}}, \mathbf{X}^{\mathrm{c}}) \mid \mathbf{X}^{\mathrm{c}}, S = 1] = \frac{\mathbb{P}(\mathbf{Z} \mid \widetilde{\mathbf{X}^{\mathrm{c}}}, S = 0)}{\mathbb{P}(\mathbf{Z} \mid \widetilde{\mathbf{X}^{\mathrm{c}}}, S = 1)}. \quad (6)$$

Since $\mathbb{P}(\mathbf{X}^{\mathrm{m}} \mid \mathbf{X}^{\mathrm{c}}, S = 0)$ is identifiable and $\mathbb{P}(\mathbf{X}^{\mathrm{m}}, S = 0)$ is available from the observed data, $\mathbb{P}(\mathbf{X}^{\mathrm{m}}, \mathbf{X}^{\mathrm{c}}, S = 0)$ is identified as

$$\mathbb{P}(\mathbf{X}^{\mathrm{m}}, \mathbf{X}^{\mathrm{c}}, S = 0) = \mathbb{P}(\mathbf{X}^{\mathrm{m}} \mid \mathbf{X}^{\mathrm{c}}, S = 0) \cdot \mathbb{P}(\mathbf{X}^{\mathrm{c}}, S = 0),$$

which is equivalent to $\mathbb{P}(\mathbf{X}^{\mathrm{m}}, \mathbf{X}^{\mathrm{c}}, R = 0)$. Therefore, $F(\mathbf{X} \mid R = 0)$ is also identifiable in Setting 1.

**Step 2.** Proof of the identifiability of $\mathbb{E}[Y(t) \mid \mathbf{X}]$.

**Step 2.1.** Proof in Setting 1 (separating set in $\mathcal{R}$).

In Setting 1, all covariates $\mathbf{X}$ in Dataset $\mathcal{R}$ are observable and include all the variables from the separating set under Assumption 3.1. Therefore, under Assumptions 2.3, 2.4, 2.5, and 2.6, $\mathbb{E}[Y(t) \mid \mathbf{X}]$ is identified by $\mathbb{E}[Y(t) \mid \mathbf{X}] = \mathbb{E}[Y \mid \mathbf{X}, T = t, R = 1]$ based on Theorem 2.9.

**Step 2.2.** Proof in Setting 2 (separating set in $\mathcal{O}$).

In Setting 2, to prove the identifiability of $\mathbb{E}[Y(t) \mid \mathbf{X}]$, we must prove the identifiability of $\mathbb{P}(\mathbf{X}, T, Y(t) \mid R = 1)$.

Under Assumption 2.3, we have $T \perp\!\!\!\perp \mathbf{X}^{\mathrm{m}} \mid R = 1$. Therefore, $\mathbb{P}(\mathbf{X}, T, Y(t) \mid R = 1)$ is identified as

$$\mathbb{P}(\mathbf{X}, T, Y(t) \mid R = 1) = \mathbb{P}(\mathbf{X}^{\mathrm{m}} \mid \mathbf{X}^{\mathrm{c}}, Y(t), R = 1)$$
$$\times \mathbb{P}(\mathbf{X}^{\mathrm{c}}, T, Y(t) \mid R = 1),$$

where $\mathbb{P}(\mathbf{X}^{\mathrm{c}}, Y(t), T \mid R = 1)$ is available in Dataset $\mathcal{R}$, and $\mathbb{P}(\mathbf{X}^{\mathrm{m}} \mid \mathbf{X}^{\mathrm{c}}, Y(t), R = 1)$ is identified by

$$\mathbb{P}(\mathbf{X}^{\mathrm{m}} \mid \mathbf{X}^{\mathrm{c}}, Y(t), R = 1) = \mathbb{P}(\mathbf{X}, R = 1)$$
$$\times \frac{\mathbb{P}(Y(t) \mid \mathbf{X}, R = 1)}{\mathbb{P}(\mathbf{X}^{\mathrm{c}}, Y(t), R = 1)}.$$

Here, $\mathbb{P}(\mathbf{X}^{\mathrm{c}}, Y(t), R = 1)$ is identifiable in Dataset $\mathcal{R}$; $\mathbb{P}(\mathbf{X}, R = 1) \equiv \mathbb{P}(\mathbf{X}, S = 0)$ (because $S = 1 - R$ in Setting 2), which is identifiable based on Lemmas 3.4 and 3.5; and $\mathbb{P}(Y(t) \mid \mathbf{X}, R = 1)$ is identifiable under Assumption 3.1, where $\mathbb{P}(Y(t) \mid \mathbf{X}, R = 1) = \mathbb{P}(Y(t) \mid \mathbf{X}, R = 0)$, which is available in Dataset $\mathcal{O}$.

Consequently, $\mathbb{P}(\mathbf{X}, T, Y(t) \mid R = 1)$ and, thus, $\mathbb{E}[Y(t) \mid \mathbf{X}]$ are identifiable in Dataset $\mathcal{R}$ based on Theorem 2.9.

The overall process of the proof is illustrated in Figure 2.

$\square$

Theorem 3.3 is sufficient for the estimation of the TATE. However, in order to estimate the TATE in a doubly robust manner, we further introduce the selection scores.

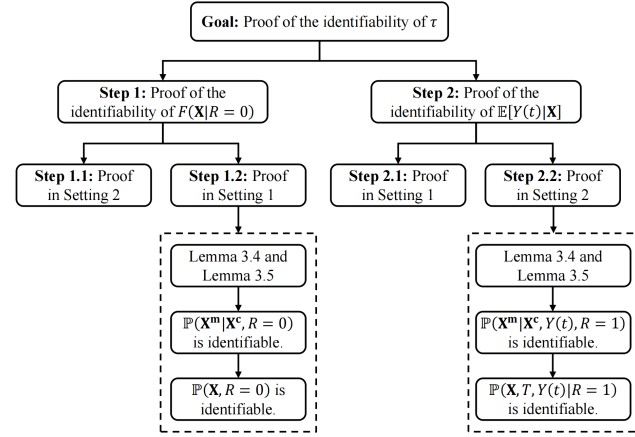

*Figure 2.* Proof process of Theorem 4.1.

**Definition 3.6. (Selection scores.)** The selection score of $S$ is $\pi_s(\mathbf{X}) \equiv \mathbb{P}(S = 1 \mid \mathbf{X})$, and the selection score of $R$ is $\pi_r(\mathbf{X}) \equiv \mathbb{P}(R = 1 \mid \mathbf{X})$.

**Corollary 3.7.** *(Identification of the selection scores.) Under Assumptions 2.3, 2.4, 2.5, 2.6, 3.1, and 3.2, $\pi_s(\mathbf{X})$ is identified as*

$$\pi_s(\mathbf{X}) = \left(1 + \widetilde{\mathrm{OR}}(\mathbf{X}^{\mathrm{m}}, \mathbf{X}^{\mathrm{c}}) \cdot \frac{\mathbb{P}(S = 0 \mid \widetilde{\mathbf{X}^{\mathrm{c}}})}{\mathbb{P}(S = 1 \mid \widetilde{\mathbf{X}^{\mathrm{c}}})}\right)^{-1},$$

*and $\pi_r(\mathbf{X})$ is identified as $\pi_r(\mathbf{X}) = \pi_s(\mathbf{X})$ in Setting 1 and $\pi_r(\mathbf{X}) = 1 - \pi_s(\mathbf{X})$ in Setting 2.*

The detailed proof of Corollary 3.7 is in Appendix F.

## 4. Two-Stage Doubly Robust TATE Estimation

Based on Theorem 3.3 and Corollary 3.7, we propose a novel Two-Stage Doubly Robust (2SDR) algorithm to estimate the TATE. 2SDR consists of two stages:

- **(Stage I.)** Shadow variable selection and imputation of the missing covariates.
- **(Stage II.)** Selection score and TATE estimation.

### 4.1. Stage I of 2SDR

#### 4.1.1. SHADOW VARIABLE SELECTION

Theorem 3.3 and Corollary 3.7 indicate that identifying the TATE and the selection score necessitates the shadow variables $\mathbf{Z}$. However, under Assumption 3.2, although $\mathbf{Z}$ is included in the common covariates $\mathbf{X}^{\mathrm{c}}$, we do not know which variables satisfy Assumption 3.2. Therefore, we first propose automatically selecting the variables from $\mathbf{X}^{\mathrm{c}}$ that satisfy Assumption 3.2, i.e., selecting $\mathbf{Z}$ from $\mathbf{X}^{\mathrm{c}}$.

Assumption 3.2 consists of two testable sub-assumptions.

Therefore, in the process of shadow variable selection, 2SDR conducts hypothesis testing on the variables in $\mathbf{X}^c$ to eliminate those that do not pass the test, leaving the variables that pass the test as the set of shadow variables.

To reduce the number of time-consuming hypothesis tests, we first apply Adaptive Lasso (Zou, 2006) for a quick initial screening of the variables in $\mathbf{X}^c$, selecting those strongly correlated with $\mathbf{X}^m$, as Assumption 3.2(1) requires that $\mathbf{Z}$ be conditionally correlated with $\mathbf{X}^m$. The set of filtered variables is denoted as $\mathbf{X}^f \subset \mathbf{X}^c$.

Next, we perform hypothesis testing for Assumption 3.2 on the filtered variables. Assumption 3.2(1) involves only the dataset where all the covariates are observable, so standard conditional independence test methods with a reject threshold $\alpha_1$ can be directly applied (Zhang et al., 2011; Sen et al., 2017; Strobl et al., 2019; Zheng et al., 2024). However, Assumption 3.2(2) involves both datasets, and the values of $\mathbf{X}^m$ are missing in one of them. Therefore, we cannot directly apply a standard conditional independence test to verify this sub-assumption, but instead, we conduct the test based on the following theorem (d'Haultfoeuille, 2010).

**Theorem 4.1.** *(Hypothesis testing for Assumption 3.2(2).) Suppose that $\mathbb{P}(\mathbf{X}^m > 0)$ holds almost surely. Then, Assumption 3.2(2), i.e., $\mathbf{Z} \perp\!\!\!\perp S \mid \mathbf{X}^m, \widetilde{\mathbf{X}^c}$, can be rejected if and only if there exists no solution to the following equation that belongs to $(0, 1]$.*

$$\mathbb{E}\left[\left(\frac{S}{Q(\mathbf{X}^m, \widetilde{\mathbf{X}^c})} - 1\right) \cdot \left(\frac{\mathbf{Z}}{\widetilde{\mathbf{X}^c}}\right)\right] = 0,$$

*where $Q$ is the unknown function to be solved for.*

The proof of Theorem 4.1 can be found in Appendix A.3 of d'Haultfoeuille (2010).

Based on Theorem 4.1, we solve for $Q$ by minimizing the following objective function:

$$\ell_Q = \frac{1}{n} \sum_{i=1}^n \left\| \left(\frac{s_i}{Q(\mathbf{x}_i^m, \widetilde{\mathbf{x}_i^f})} - 1\right) \cdot \left(\frac{\mathbf{z}_i^c}{\widetilde{\mathbf{x}_i^f}}\right) \right\|,$$

where $Q$ is constrained to the range $(0, 1]$ by an activation function, such as the sigmoid function, $\mathbf{Z}^c \subset \mathbf{X}^f$ denotes the candidate shadow variables, and $\widetilde{\mathbf{X}^f}$ denotes the set difference $\mathbf{X}^c \setminus \mathbf{Z}^c$. If $\ell_Q$ converges to a value less than a threshold $\alpha_2$, the candidate shadow variables pass the test.

To ensure that the completeness condition for the shadow variables is satisfied, we aim to select $d$ shadow variables from $\mathbf{X}^f$, where $d$ equals the number of variables in $\mathbf{X}^m$ (Miao et al., 2024). Therefore, we iterate through $\mathbf{X}^f$, selecting a candidate shadow variable in each iteration and performing hypothesis testing. The loop continues until $d$ candidates pass the test, at which point the corresponding shadow variables are selected, and the loop terminates.

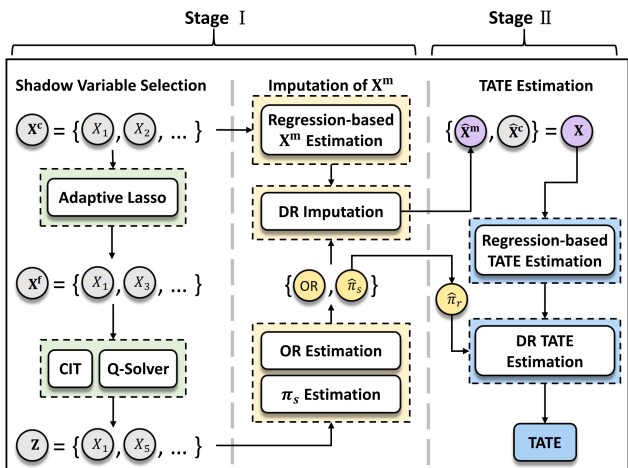

*Figure 3.* Overview of 2SDR.

The time complexity of the shadow variable selection process depends on the specific method used to test Assumption 3.2(1), which, in our implementation, is the Randomized Conditional Independence Test (RCIT) (Strobl et al., 2019). As a result, the time complexity of this process is $O(np^2)$, where $p$ is the number of variables in $\mathbf{X}$. A detailed time complexity analysis can be found in Appendix I.

### 4.1.2. IMPUTATION OF THE MISSING COVARIATES

Through the shadow variable selection process, we obtain the set of shadow variables $\mathbf{Z}$. Next, based on Theorem 3.3 and Corollary 3.7, we use $\mathbf{Z}$ to impute the missing values of $\mathbf{X}^m$ in a doubly robust manner, as detailed below.

First, we estimate the odds ratio function using Equations (3) and (4). Specifically, we begin by employing kernel density estimation (Silverman, 2018) to estimate $f(\mathbf{Z} \mid \widetilde{\mathbf{X}^c}, S = 1)$ and $f(\mathbf{Z} \mid \widetilde{\mathbf{X}^c}, S = 0)$, where $f(\mathbf{Z} \mid \widetilde{\mathbf{X}^c}, S)$ is the probability density function of $\mathbf{Z}$ conditional on $\widetilde{\mathbf{X}^c}$ and $S$. Then, we estimate $\widetilde{\mathrm{OR}}(\mathbf{X}^m, \mathbf{X}^c)$ by minimizing the following objective function:

$$\ell_{\widetilde{\mathrm{OR}}} = \frac{1}{n} \sum_{i:s_i=1} \left( \widetilde{\mathrm{OR}}(\mathbf{x}_i^m, \mathbf{x}_i^c) - \frac{\hat{f}(\mathbf{z}_i \mid \widetilde{\mathbf{x}^c}_i, s_i = 0)}{\hat{f}(\mathbf{z}_i \mid \widetilde{\mathbf{x}^c}_i, s_i = 1)} \right)^2,$$

where $\hat{f}(\mathbf{Z} \mid \widetilde{\mathbf{X}^c}, S)$ denotes the estimated probability density functions. Subsequently, we obtain the estimated odds ratio function $\widehat{\mathrm{OR}}(\mathbf{X}^m, \mathbf{X}^c)$ using Equation (4).

Next, we learn a function $\psi(\mathbf{X}^c)$ to estimate $\mathbb{E}[\mathbf{X}^m \mid \mathbf{X}^c, S = 0]$ using a doubly robust method (Miao & Tchetgen Tchetgen, 2016; Kennedy, 2023) by minimizing the

following objective function:

$$\ell_\psi = \frac{1}{n} \sum_{i=1}^{n} \left( \psi(\mathbf{x}_i^c) - \hat{\mathbf{x}}_i^m \right)^2 ,$$

where $\hat{\mathbf{x}}_i^m$ denotes the doubly robust estimate of $\mathbb{E}[\mathbf{X}^m \mid \mathbf{X}^c = \mathbf{x}_i^c, S = 0]$, detailed as follows.

$$\hat{\mathbf{x}}_i^m = s_i \cdot \hat{w}_i \cdot \left( \mathbf{x}_i^m - \hat{\delta}(\mathbf{x}_i^c) \right) + \hat{\delta}(\mathbf{x}_i^c).$$

Here, $w_i = 1/\hat{\pi}_s(\mathbf{x}_i)$, where $\hat{\pi}_s$ is the estimate of $\pi_s$, and $\hat{\delta}$ is the regression-based estimate of $\mathbb{E}[\mathbf{X}^m \mid \mathbf{X}^c = \mathbf{x}_i^c, S = 0]$.

Based on Corollary 3.7, $\hat{\pi}_s$ is obtained by

$$\hat{\pi}_s(\mathbf{X}) = \left( 1 + \widetilde{\mathrm{OR}}(\mathbf{X}^m, \mathbf{X}^c) \cdot \frac{1 - \hat{\gamma}(\widetilde{\mathbf{X}^c})}{\hat{\gamma}(\widetilde{\mathbf{X}^c})} \right)^{-1} ,$$

where $\hat{\gamma}$ is the estimate of $\mathbb{P}(S = 1 \mid \widetilde{\mathbf{X}^c})$, obtained by minimizing the following objective function:

$$\ell_\gamma = -\frac{1}{n} \sum_{i=1}^{n} (1 - s_i) \cdot \log(1 - \gamma(\widetilde{\mathbf{x}_i^c})) + s_i \cdot \log(\gamma(\widetilde{\mathbf{x}_i^c})).$$

Based on Equation (2), $\hat{\delta}$ is obtained by minimizing the following objective function:

$$\ell_\delta = \frac{1}{n} \sum_{i:s_i=1} \left( \delta(\mathbf{x}_i^c) - \frac{\mathrm{OR}\left(\mathbf{x}_i^m, \mathbf{x}_i^c\right) \cdot \hat{\theta}\left(\mathbf{x}_i^c\right)}{\mathrm{OR}\left(\hat{\theta}\left(\mathbf{x}_i^c\right), \mathbf{x}_i^c\right)} \right)^2 ,$$

where $\hat{\theta}$ is the estimate of $\mathbb{E}[\mathbf{X}^m \mid \mathbf{X}^c, S = 1]$, obtained by minimizing the following objective function:

$$\ell_\theta = \frac{1}{n} \sum_{i:s_i=1} \left( \theta(\mathbf{x}_i^c) - \mathbf{x}_i^m \right)^2 .$$

Consequently, we can use the estimate of $\mathbb{E}[\mathbf{X}^m \mid \mathbf{X}^c, S = 0]$, i.e., $\hat{\psi}(\mathbf{X}^c)$ to impute the missing values of $\mathbf{X}^m$.

The proposed doubly robust imputation model is consistent, as guaranteed by the following theorem.

**Theorem 4.2.** *Under Assumptions 2.3, 2.4, 2.5, 2.6, 3.1, and 3.2, the doubly robust imputation model $\hat{\psi}$ is consistent if the $\widetilde{\mathrm{OR}}(\mathbf{X}^m, \mathbf{X}^c)$ model is correctly specified and either the regression model of the missing covariates $\mathbf{X}^m$ or the selection score model of $S$ is correctly specified.*

The proof of Theorem 4.2 is provided in Appendix G.

### 4.2. Stage II of 2SDR

In the first stage, we impute the missing values of $\mathbf{X}^m$, making all the covariates in both datasets fully observable.

This enables us to perform a doubly robust estimation of the TATE (Dahabreh & Hernán, 2019; Dahabreh et al., 2019).

$$\hat{\tau} = \frac{1}{n_{\mathcal{R}}} \sum_{i=1}^{n_{\mathcal{R}}} \frac{n_{\mathcal{R}}}{n_{\mathcal{O}}} \frac{1 - \hat{\pi}_r(\mathbf{x}_i)}{\hat{\pi}_r(\mathbf{x}_i)} \left( \frac{t_i \cdot (y_i - \hat{\mu}_1(\mathbf{x}_i))}{\zeta(\mathbf{x}_i)} \right)$$
$$- \frac{1}{n_{\mathcal{R}}} \sum_{i=1}^{n_{\mathcal{R}}} \frac{n_{\mathcal{R}}}{n_{\mathcal{O}}} \frac{1 - \hat{\pi}_r(\mathbf{x}_i)}{\hat{\pi}_r(\mathbf{x}_i)} \left( \frac{(1 - t_i) \cdot (y_i - \hat{\mu}_0(\mathbf{x}_i))}{1 - \zeta(\mathbf{x}_i)} \right)$$
$$+ \frac{1}{n_{\mathcal{O}}} \sum_{i=1}^{n_{\mathcal{O}}} \left( \hat{\mu}_1(\mathbf{x}_i) - \hat{\mu}_0(\mathbf{x}_i) \right) ,$$

Here, $\zeta(\mathbf{X}) = \mathbb{P}(T \mid \mathbf{X})$ is the propensity score, which is a constant under Assumption 2.3; $\hat{\pi}_r$ is the estimate of $\pi_r$ and can be directly obtained from the first stage, i.e., $\hat{\pi}_r = \hat{\pi}_s$ for Setting 1 and $\hat{\pi}_r = 1 - \hat{\pi}_s$ for Setting 2; and $\hat{\mu}_0$ and $\hat{\mu}_1$ are the estimates of $\mathbb{E}[Y \mid \mathbf{X}, T = 0, S = 1]$ and $\mathbb{E}[Y \mid \mathbf{X}, T = 1, S = 1]$, respectively, obtained by minimizing the following objective functions:

$$\ell_{\mu_0} = \frac{1}{n_{\mathcal{R}}} \sum_{i=1}^{n_{\mathcal{R}}} \left( (1 - t_i) \cdot (\mu_0(\mathbf{x}_i) - y_i) \right)^2 ,$$

$$\ell_{\mu_1} = \frac{1}{n_{\mathcal{R}}} \sum_{i=1}^{n_{\mathcal{R}}} \left( t_i \cdot (\mu_1(\mathbf{x}_i) - y_i) \right)^2 .$$

The proposed doubly robust TATE estimation model is consistent, as guaranteed by the following theorem.

**Theorem 4.3.** *Under Assumptions 2.3, 2.4, 2.5, 2.6, 3.1, and 3.2, if the imputation model is consistent, the TATE estimator $\hat{\tau}$ is consistent if the $\widetilde{\mathrm{OR}}$ model is correctly specified and either the regression model of the outcome $Y$ or the selection score model of $S$ is correctly specified.*

The proof of Theorem 4.3 is provided in Appendix H.

## 5. Experiments

### 5.1. Experimental Setup

In our experiments, we compared the proposed method with the following baselines: (1) The average treatment effect obtained directly from the RCT data (RCT), (2) Inverse Probability of Sampling Weighting (IPSW) (Cole & Stuart, 2010), (3) Calibration Weighting (CW) (Hartman et al., 2015), (4) G-formula (Dahabreh et al., 2019), (5) Augmented IPSW (AIPSW) (Dahabreh et al., 2019), and (6) Augmented CW (ACW) (Lee et al., 2023), along with their versions incorporating data imputation (Hughes et al., 2019; Mayer et al., 2023), denoted by their name with the 'Imp' subscript.

We conducted experiments on both synthetic and real-world datasets, with each experiment repeated 20 times. We report the mean and standard deviation (std) of the Mean Absolute Error (MAE) between the TATE estimates and

*Table 1.* TATE estimation results (MAE) on the low-dimensional synthetic dataset (mean±std), with bold values indicating the best performance.

| METHODS | $n_{\mathcal{R}} = 2000, n_{\mathcal{O}} = 10000$ | | $n_{\mathcal{R}} = 1000, n_{\mathcal{O}} = 10000$ | | $n_{\mathcal{R}} = 500, n_{\mathcal{O}} = 10000$ | |
| | SETTING 1 | SETTING 2 | SETTING 1 | SETTING 2 | SETTING 1 | SETTING 2 |
|---|---|---|---|---|---|---|
| RCT | 1.640±1.084 | 1.751±0.913 | 1.685±1.047 | 1.803±1.223 | 2.260±1.313 | 2.037±1.461 |
| IPSW | 6.460±0.301 | 6.564±0.375 | 7.034±0.212 | 6.842±0.129 | 6.999±0.152 | 7.066±0.167 |
| CW | 2.062±1.551 | 2.197±1.075 | 2.566±1.860 | 2.789±1.215 | 4.982±2.765 | 4.540±2.154 |
| G-FORMULA | 0.663±0.260 | 0.659±0.266 | 0.720±0.263 | 0.692±0.377 | 1.145±0.814 | 0.834±0.710 |
| AIPSW | 0.663±0.261 | 0.660±0.265 | 0.720±0.264 | 0.692±0.377 | 1.145±0.815 | 0.834±0.710 |
| ACW | 0.662±0.260 | 0.659±0.266 | 0.720±0.263 | 0.693±0.377 | 1.145±0.814 | 0.833±0.709 |
| $\text{IPSW}_{\text{Imp}}$ | 6.350±0.159 | 6.458±0.188 | 6.829±0.126 | 6.887±0.194 | 6.910±0.101 | 6.924±0.103 |
| $\text{CW}_{\text{Imp}}$ | 2.020±1.569 | 2.006±1.020 | 2.410±1.235 | 2.453±0.917 | 3.794±3.436 | 3.108±2.322 |
| $\text{G-FORMULA}_{\text{Imp}}$ | 0.539±0.305 | 0.815±0.310 | 0.614±0.355 | 0.998±0.450 | 0.961±0.625 | 1.787±0.946 |
| $\text{AIPSW}_{\text{Imp}}$ | 0.539±0.303 | 0.829±0.306 | 0.614±0.355 | 1.084±0.453 | 0.960±0.625 | 1.792±0.949 |
| $\text{ACW}_{\text{Imp}}$ | 0.539±0.306 | 0.816±0.310 | 0.614±0.356 | 0.998±0.451 | 0.961±0.625 | 1.789±0.947 |
| 2SDR | **0.268±0.212** | **0.284±0.224** | **0.423±0.264** | **0.431±0.296** | **0.533±0.387** | **0.489±0.327** |

the ground truth values. In each experiment, we randomly split the dataset into training, validation, and test sets with a 60/20/20 ratio. Implementation details of the proposed method, as well as the software and hardware we used, are provided in Appendix J.

The source code of 2SDR is available at https://github.com/ZJUBaohongLi/2SDR.

### 5.2. Experiments on Synthetic Datasets

#### 5.2.1. DATASETS

To evaluate the effectiveness of the proposed method in both cases, we generated datasets based on Setting 1 to simulate the case where the values of $\mathbf{X}^m$ are missing in Dataset $\mathcal{O}$ and Setting 2 to simulate the case where the values of $\mathbf{X}^m$ are missing in Dataset $\mathcal{R}$. The detailed data generation process is in Appendix K.

In general, due to the high cost of RCTs, $n_{\mathcal{R}}$ is typically much smaller than $n_{\mathcal{O}}$ (Kohavi & Longbotham, 2011; Kallus et al., 2018). Therefore, in order to evaluate the robustness of the proposed method with small-scale RCT data, we fixed $n_{\mathcal{O}}$ to 10000 while setting $n_{\mathcal{R}} = \{500, 1000, 2000\}$ to compare the performance of 2SDR with the baselines across different RCT scales.

To evaluate the performance of the proposed method in the case of a larger number of covariates, we also generated a high-dimensional dataset. The detailed data generation process can be found in Appendix K.

#### 5.2.2. RESULTS

We report the TATE estimation results under different RCT scales in Table 1, with the following observations and conclusions: (1) As the RCT scale decreases, the TATE estimation error increases for all methods. However, 2SDR always

achieves the best performance across all RCT scales, demonstrating its robustness. (2) In Setting 1, the performance of the baselines using imputation is better than that without imputation. However, in Setting 2, the regression-based baselines, including doubly robust ones, fail to produce better estimates using imputation. This observation can be attributed to the fact that, in Setting 1, the regression process involves the fully observed covariates from the RCT data, and the imputation error only affects the prediction based on the imputed covariates from the observational data. In contrast, in Setting 2, the regression process involves imputed covariate values from the RCT data, accumulating imputation errors, and introducing bias into the regression model. On the other hand, 2SDR performs well in both settings because it selects shadow variables and uses them to obtain unbiased imputation. (3) The doubly robust baselines do not perform better, as both the selection scores and outcome estimates are biased when variables in the separating set are missing. In contrast, 2SDR ensures that both the selection score and outcome estimate are unbiased, leading to its superior performance.

Additional experimental results, including results on the high-dimensional dataset, a comparison of the imputation accuracy between the proposed method and the baselines using imputation, the hyperparameter analysis, and the robustness analysis of the proposed method when Assumption 3.2 is violated, are presented in Appendix L.

### 5.3. Experiments on Real-World Datasets

#### 5.3.1. DATASETS

To validate the effectiveness of the proposed method in real-world applications, we conducted experiments on two real datasets, the AIDS Clinical Trial Group (ACTG) and Jobs Training Partnership Act (JTPA) datasets, with the ACTG

*Table 2.* TATE estimation results (MAE) on real-world datasets (mean$\pm$std scaled by $10^3$), with bold values indicating the best performance.

| METHODS | ACTG | JTPA |
|---|---|---|
| RCT | 0.032$\pm$0.016 | 1.366$\pm$0.709 |
| IPSW | 640.4$\pm$98.80 | 1.064$\pm$0.519 |
| CW | 0.367$\pm$0.044 | 9.749$\pm$0.759 |
| G-FORMULA | 0.035$\pm$0.024 | 2.247$\pm$0.965 |
| AIPSW | 27.91$\pm$15.78 | 2.249$\pm$0.966 |
| ACW | 0.035$\pm$0.024 | 2.245$\pm$0.966 |
| IPSW$_{\text{Imp}}$ | 656.5$\pm$159.1 | 1.230$\pm$0.688 |
| CW$_{\text{Imp}}$ | 0.340$\pm$0.036 | 8.904$\pm$1.305 |
| G-FORMULA$_{\text{Imp}}$ | 0.037$\pm$0.026 | 2.229$\pm$1.553 |
| AIPSW$_{\text{Imp}}$ | 14.97$\pm$6.543 | 2.230$\pm$1.554 |
| ACW$_{\text{Imp}}$ | 0.037$\pm$0.026 | 2.229$\pm$1.553 |
| 2SDR | **0.011$\pm$0.010** | **0.992$\pm$0.490** |

dataset corresponding to Setting 1 and the JTPA dataset to Setting 2, detailed as follows.

**The ACTG dataset.** The RCT data in this dataset is from the ACTG 175, which aimed to investigate the treatment effect of a specific drug on the CD4 count of AIDS patients. The dataset contains covariates such as age, gender, weight, race, and pre-treatment CD4 count (Hammer et al., 1997). Following Cole & Stuart (2010); Dahabreh et al. (2023), we aim to generalize the results from the ACTG 175 trial to the target population of individuals infecting HIV in the United States in 2006 (Prejean et al., 2008; Hall et al., 2008). In contrast to the ACTG 175, the age and pre-treatment CD4 count are missing in the target population, aligning with Setting 1. Since there is no ground-truth TATE in the original observational data from the target population, we cannot evaluate the performance of the TATE estimation results. Therefore, we randomly selected 558 individuals from the ACTG 175 who were either under 30 or over 50 years old and omitted their age, pre-treatment CD4 count, and post-treatment CD4 count to serve as the observational data from the target population, with the remaining 1581 individuals forming the RCT data.

**The JTPA dataset.** The national JTPA study, which was conducted from 1987 to 1989, aimed to assess the effectiveness of job training programs in helping individuals find employment and increase their earnings (Doolittle & Traeger, 1990; Bloom et al., 1993). Following Huang (2024), we selected data from the site of Coosa Valley, Georgia, from the original JTPA trial as the RCT data, with their years of education and high school or equivalency diploma status omitted. The data from the remaining 15 sites were used as the observational data from the target population. The RCT data contains 788 individuals, and the observational data contains 5314 individuals.

### 5.3.2. RESULTS

We report the TATE estimation results on the two real-world datasets in Table 2, with the following observations and conclusions: (1) On the ACTG dataset, the performance of all baselines is even worse than using the ATE obtained directly from the RCT data. This is because the pre-treatment CD4 count, which significantly impacts the outcome, is missing in the observational data from the target population. As a result, methods that fail to impute the missing pre-treatment CD4 count values accurately are highly biased. (2) Similarly, on the JTPA dataset, due to the absence of the years of education in the RCT data, which significantly impacts the outcome, the majority of the baseline models exhibit substantial bias when estimating the TATE, performing even worse than using the ATE obtained directly from the RCT data. (3) 2SDR achieves the best performance on both real-world datasets, demonstrating its practical value in real-world scenarios.

## 6. Conclusion

In this paper, we focus on the problem of generalizing treatment effects from RCTs to target populations across different environments. To tackle the challenges of selection bias and missing covariates posed by environmental shifts, we propose a novel TATE identifiability framework that relaxes the assumptions made in prior work. Based on this theoretical foundation, we propose a novel doubly robust TATE estimation method, 2SDR, which effectively addresses these challenges. Extensive experimental results on synthetic and real-world datasets demonstrate the effectiveness of the proposed method and its practical value in real-world applications. The main limitation of 2SDR is that when the dimension of the covariates is extremely high, the shadow variable selection process in the first stage becomes time-consuming. Additionally, if the number of the partially observable covariates exceeds that of the common covariates shared by both data, it becomes challenging to ensure the validity of Assumption 3.2. In future work, we will explore using dimensionality reduction methods, such as representation learning, to address the above limitation.

## Acknowledgements

This work was supported in part by the National Key Research and Development Program of China (2024YFE0203700), National Natural Science Foundation of China (62376243, 62441605), and "Pioneer" and "Leading Goose" R&D Program of Zhejiang (2025C02037). All opinions in this paper are those of the authors and do not necessarily reflect the views of the funding agencies.

## Impact Statement

This paper presents work whose goal is to advance the field of Machine Learning. In this paper, we report experimental results on two real-world datasets to evaluate the performance of our method in practical scenarios. However, the results should not be interpreted as providing direct guidance for treatment assignments in the corresponding real-world applications.

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

# A. Related Work

RCTs have been widely used to study the effects of new interventions in a wide range of disciplines (Fisher, 1971; Wu & Hamada, 2011; Xiong et al., 2024a;b). Recently, many studies have aimed to generalize treatment effects from RCTs to target populations by combining RCT data with observational data from the target population to estimate the TATE. Prior work show that the TATE can be identified by a separating set (Cole & Stuart, 2010; Tipton, 2013; Kern et al., 2016; Egami & Hartman, 2021; Pearl & Bareinboim, 2022). The separating set is a set of variables affecting both treatment effect heterogeneity and environmental shifts. Many approaches are proposed to adjust for the separating set through different technologies for generalizing treatment effects from RCTs to target populations (Stuart et al., 2011; Pearl & Bareinboim, 2011; O'Muircheartaigh & Hedges, 2014; Hartman et al., 2015; Lesko et al., 2017; Buchanan et al., 2018; Dahabreh & Hernán, 2019; Dahabreh et al., 2019; Lee et al., 2023; Li et al., 2024a). For example, Cole & Stuart (2010); Stuart et al. (2011) proposed first estimating the selection score, i.e., $\mathbb{P}(R = 1 \mid \mathbf{X})$, and then using it to reweight the RCT samples in order to adjust for $\mathbf{X}$. Hartman et al. (2015) proposed the Calibration Weighting (CW) method, which learns a sample weight matrix that constrains the covariate distribution in the RCT data to align with that of the observational data from the target population. The sample weight matrix simulates the selection score and avoids issues arising from model misspecification. Dahabreh & Hernán (2019); Dahabreh et al. (2019) proposed a regression-based method, G-formula, and a doubly robust method, Augmented IPSW (IPSW), to adjust for the covariates more effectively. Lee et al. (2023) also proposed a doubly robust version of CW, Augmented CW (ACW), to enhance the robustness of the estimation.

All the above methods assume that the covariates shared by both groups contain the separating set. However, in real-world scenarios under environmental shifts, it is challenging to ensure that the covariates collected in both groups are exactly the same, leading to the missing covariates problem—where certain covariates present in one group are entirely absent in the other. If the missing covariates include variables from the separating set, previous methods may fail to perform as expected. Recently, several sensitivity analysis methods were proposed to detect whether the fully observed covariates contain all the variables from the separation set (Nguyen et al., 2017; Andrews & Oster, 2019; Colnet et al., 2022; Dahabreh et al., 2023; Huang, 2024). However, to the best of our knowledge, no existing work can identify and estimate the TATE when the separating set is observable in only one of the two datasets.

This work also complements the recent literature on treatment effect estimation and parameter inference for panel data with nonrandom missingness (Xiong & Pelger, 2023; Duan et al., 2024a). In our setting, covariates form partially observed matrices in either the RCT or observational data, and the missingness is affected by experimental shifts. This is closely related to Duan et al. (2024b), which transfers information across panels under general missingness–conceptually similar to our use of 2SDR to transfer information from RCT to observational data.

# B. Discussion on the Scenarios of Environmental Shifts Applicable to This Paper

The applicable scenarios of environmental shifts for this paper can be summarized as follows:

- **Shifts in covariates.** As stated in Definition 2.1, the original definition of environmental shifts in this paper is covariate shifts, i.e., $\mathbb{P}(\mathbf{X} \mid R = 1) \neq \mathbb{P}(\mathbf{X} \mid R = 0)$.
- **Shifts in both the outcome and covariates.** Although this is not explicitly stated in Definition 2.1, due to the fact that some covariates may be causes of the outcome, the distribution of the outcome will also shift along with the distribution of such covariates, i.e., $\mathbb{P}(Y \mid R = 1) \neq \mathbb{P}(Y \mid R = 0)$, or alternatively, $\mathbb{P}(\mathbf{X}, Y \mid R = 1) \neq \mathbb{P}(\mathbf{X}, Y \mid R = 0)$.
- **Shifts in the conditional distribution of the outcome.** Given that $Y(t) \per\!\!\!\perp R \mid \mathbf{X}$ holds under Assumption 3.1, we have $\mathbb{P}(Y \mid \mathbf{X}, R = 1) = \mathbb{P}(Y \mid \mathbf{X}, R = 0)$. Therefore, our problem setting essentially assumes the absence of $Y \mid \mathbf{X}$ shifts. However, for the common covariates $\mathbf{X}^c$ shared by the two datasets, there still exist $Y \mid \mathbf{X}^c$ shifts, i.e., $\mathbb{P}(Y \mid \mathbf{X}^c, R = 1) \neq P(Y \mid \mathbf{X}^c, R = 0)$. Assumption 3.2 required by the proposed 2SDR is still satisfied under $Y \mid \mathbf{X}$ shifts, and therefore, 2SDR can address $Y \mid \mathbf{X}^c$ shifts. However, existing methods rely on Assumption 2.8, which requires that $Y(t) \per\!\!\!\perp R \mid \mathbf{X}^c$ holds. Under $Y \mid \mathbf{X}^c$ shifts, Assumption 2.8 does not hold, and thus, in contrast to 2SDR, previous methods cannot address $Y \mid \mathbf{X}^c$ shifts.

# C. Proof of Theorem 2.9

*Proof.* Under Assumptions 2.3, 2.4, and 2.6, we have $\mathbb{E}[Y(t) \mid \mathbf{W}, R = 1] = \mathbb{E}[Y \mid \mathbf{W}, T = t, R = 1]$.

From Definition 2.7, we obtain $\mathbb{E}[Y \mid \mathbf{W}, T, R = 1] = \mathbb{E}[Y \mid \mathbf{W}, T] = \mathbb{E}[Y \mid \mathbf{W}, T, R = 1]$ under Assumption 2.5.

Therefore, based on Definition 2.2, we have

$$\tau = \int \left( \mathbb{E}[Y(1) \mid \mathbf{W} = \mathbf{w}, R = 0] - \mathbb{E}[Y(0) \mid \mathbf{W} = \mathbf{w}, R = 0] \right) dF(\mathbf{W} = \mathbf{w} \mid R = 0)$$

$$= \int \left( \mathbb{E}[Y(1) \mid \mathbf{W} = \mathbf{w}] - \mathbb{E}[Y(0) \mid \mathbf{W} = \mathbf{w}] \right) dF(\mathbf{W} = \mathbf{w} \mid R = 0).$$

□

## D. Proof of Lemma 3.4

*Proof.* First, substituting Equation (3) into Equation (2) yields

$$\mathbb{P}(\mathbf{X}^{\mathrm{m}} \mid \mathbf{X}^{\mathrm{c}}, S = 0) = \frac{\frac{\mathbb{P}(S=0 \mid \mathbf{X}^{\mathrm{m}}, \mathbf{X}^{\mathrm{c}})}{\mathbb{P}(S=1 \mid \mathbf{X}^{\mathrm{m}}, \mathbf{X}^{\mathrm{c}})} \cdot \frac{\mathbb{P}(S=1 \mid \mathbf{X}^{\mathrm{m}}=\mathbf{0}, \mathbf{X}^{\mathrm{c}})}{\mathbb{P}(S=0 \mid \mathbf{X}^{\mathrm{m}}=\mathbf{0}, \mathbf{X}^{\mathrm{c}})} \cdot \mathbb{P}(\mathbf{X}^{\mathrm{m}} \mid \mathbf{X}^{\mathrm{c}}, S = 1)}{\mathbb{E}\left[ \frac{\mathbb{P}(S=0 \mid \mathbf{X}^{\mathrm{m}}, \mathbf{X}^{\mathrm{c}})}{\mathbb{P}(S=1 \mid \mathbf{X}^{\mathrm{m}}, \mathbf{X}^{\mathrm{c}})} \cdot \frac{\mathbb{P}(S=1 \mid \mathbf{X}^{\mathrm{m}}=\mathbf{0}, \mathbf{X}^{\mathrm{c}})}{\mathbb{P}(S=0 \mid \mathbf{X}^{\mathrm{m}}=\mathbf{0}, \mathbf{X}^{\mathrm{c}})} \middle| \mathbf{X}^{\mathrm{c}}, S = 1 \right]}.$$

Under Assumptions 2.5 and 3.2, we have

$$\mathbb{E}\left[ \frac{\mathbb{P}(S = 0 \mid \mathbf{X}^{\mathrm{m}}, \mathbf{X}^{\mathrm{c}})}{\mathbb{P}(S = 1 \mid \mathbf{X}^{\mathrm{m}}, \mathbf{X}^{\mathrm{c}})} \cdot \frac{\mathbb{P}(S = 1 \mid \mathbf{X}^{\mathrm{m}} = \mathbf{0}, \mathbf{X}^{\mathrm{c}})}{\mathbb{P}(S = 0 \mid \mathbf{X}^{\mathrm{m}} = \mathbf{0}, \mathbf{X}^{\mathrm{c}})} \middle| \mathbf{X}^{\mathrm{c}}, S = 1 \right]$$

$$= \sum_{\mathbf{x}^{\mathrm{m}}} \frac{\mathbb{P}(S = 0 \mid \mathbf{X}^{\mathrm{m}} = \mathbf{x}^{\mathrm{m}}, \mathbf{X}^{\mathrm{c}})}{\mathbb{P}(S = 1 \mid \mathbf{X}^{\mathrm{m}} = \mathbf{x}^{\mathrm{m}}, \mathbf{X}^{\mathrm{c}})} \cdot \frac{\mathbb{P}(S = 1 \mid \mathbf{X}^{\mathrm{m}} = \mathbf{0}, \mathbf{X}^{\mathrm{c}})}{\mathbb{P}(S = 0 \mid \mathbf{X}^{\mathrm{m}} = \mathbf{0}, \mathbf{X}^{\mathrm{c}})} \cdot \mathbb{P}(\mathbf{X}^{\mathrm{m}} = \mathbf{x}^{\mathrm{m}} \mid \mathbf{X}^{\mathrm{c}}, S = 1)$$

$$= \sum_{\mathbf{x}^{\mathrm{m}}} \frac{\mathbb{P}(S = 0 \mid \mathbf{X}^{\mathrm{m}} = \mathbf{x}^{\mathrm{m}}, \mathbf{X}^{\mathrm{c}})}{\mathbb{P}(S = 1 \mid \mathbf{X}^{\mathrm{m}} = \mathbf{x}^{\mathrm{m}}, \mathbf{X}^{\mathrm{c}})} \cdot \frac{\mathbb{P}(S = 1 \mid \mathbf{X}^{\mathrm{m}} = \mathbf{0}, \mathbf{X}^{\mathrm{c}})}{\mathbb{P}(S = 0 \mid \mathbf{X}^{\mathrm{m}} = \mathbf{0}, \mathbf{X}^{\mathrm{c}})} \cdot \frac{\mathbb{P}(S = 1 \mid \mathbf{X}^{\mathrm{m}} = \mathbf{x}^{\mathrm{m}}, \mathbf{X}^{\mathrm{c}})}{\mathbb{P}(S = 1 \mid \mathbf{X}^{\mathrm{c}})} \cdot \mathbb{P}(\mathbf{X}^{\mathrm{m}} = \mathbf{x}^{\mathrm{m}} \mid \mathbf{X}^{\mathrm{c}})$$

$$= \frac{\mathbb{P}(S = 1 \mid \mathbf{X}^{\mathrm{m}} = \mathbf{0}, \mathbf{X}^{\mathrm{c}})}{\mathbb{P}(S = 0 \mid \mathbf{X}^{\mathrm{m}} = \mathbf{0}, \mathbf{X}^{\mathrm{c}}) \cdot \mathbb{P}(S = 1 \mid \mathbf{X}^{\mathrm{c}})} \cdot \sum_{\mathbf{x}^{\mathrm{m}}} \mathbb{P}(S = 0 \mid \mathbf{X}^{\mathrm{m}} = \mathbf{x}^{\mathrm{m}}, \mathbf{X}^{\mathrm{c}}) \cdot \mathbb{P}(\mathbf{X}^{\mathrm{m}} = \mathbf{x}^{\mathrm{m}} \mid \mathbf{X}^{\mathrm{c}})$$

$$= \frac{\mathbb{P}(S = 1 \mid \mathbf{X}^{\mathrm{m}} = 0, \mathbf{X}^{\mathrm{c}})}{\mathbb{P}(S = 0 \mid \mathbf{X}^{\mathrm{m}} = 0, \mathbf{X}^{\mathrm{c}}) \cdot \mathbb{P}(S = 1 \mid \mathbf{X}^{\mathrm{c}})} \cdot \sum_{\mathbf{x}^{\mathrm{m}}} \mathbb{P}(S = 0, \mathbf{X}^{\mathrm{m}} = \mathbf{x}^{\mathrm{m}} \mid \mathbf{X}^{\mathrm{c}})$$

$$= \frac{\mathbb{P}(S = 1 \mid \mathbf{X}^{\mathrm{m}} = \mathbf{0}, \mathbf{X}^{\mathrm{c}})}{\mathbb{P}(S = 0 \mid \mathbf{X}^{\mathrm{m}} = \mathbf{0}, \mathbf{X}^{\mathrm{c}})} \cdot \frac{\mathbb{P}(S = 0 \mid \mathbf{X}^{\mathrm{c}})}{\mathbb{P}(S = 1 \mid \mathbf{X}^{\mathrm{c}})}.$$

Therefore, we have

$$\frac{\frac{\mathbb{P}(S=0 \mid \mathbf{X}^{\mathrm{m}}, \mathbf{X}^{\mathrm{c}})}{\mathbb{P}(S=1 \mid \mathbf{X}^{\mathrm{m}}, \mathbf{X}^{\mathrm{c}})} \cdot \frac{\mathbb{P}(S=1 \mid \mathbf{X}^{\mathrm{m}}=\mathbf{0}, \mathbf{X}^{\mathrm{c}})}{\mathbb{P}(S=0 \mid \mathbf{X}^{\mathrm{m}}=\mathbf{0}, \mathbf{X}^{\mathrm{c}})} \cdot \mathbb{P}(\mathbf{X}^{\mathrm{m}} \mid \mathbf{X}^{\mathrm{c}}, S = 1)}{\mathbb{E}\left[ \frac{\mathbb{P}(S=0 \mid \mathbf{X}^{\mathrm{m}}, \mathbf{X}^{\mathrm{c}})}{\mathbb{P}(S=1 \mid \mathbf{X}^{\mathrm{m}}, \mathbf{X}^{\mathrm{c}})} \cdot \frac{\mathbb{P}(S=1 \mid \mathbf{X}^{\mathrm{m}}=\mathbf{0}, \mathbf{X}^{\mathrm{c}})}{\mathbb{P}(S=0 \mid \mathbf{X}^{\mathrm{m}}=\mathbf{0}, \mathbf{X}^{\mathrm{c}})} \middle| \mathbf{X}^{\mathrm{c}}, S = 1 \right]}$$

$$= \frac{\frac{\mathbb{P}(S=0 \mid \mathbf{X}^{\mathrm{m}}, \mathbf{X}^{\mathrm{c}})}{\mathbb{P}(S=1 \mid \mathbf{X}^{\mathrm{m}}, \mathbf{X}^{\mathrm{c}})} \cdot \frac{\mathbb{P}(S=1 \mid \mathbf{X}^{\mathrm{m}}=\mathbf{0}, \mathbf{X}^{\mathrm{c}})}{\mathbb{P}(S=0 \mid \mathbf{X}^{\mathrm{m}}=\mathbf{0}, \mathbf{X}^{\mathrm{c}})} \cdot \mathbb{P}(\mathbf{X}^{\mathrm{m}} \mid \mathbf{X}^{\mathrm{c}}, S = 1)}{\frac{\mathbb{P}(S=1 \mid \mathbf{X}^{\mathrm{m}}=\mathbf{0}, \mathbf{X}^{\mathrm{c}})}{\mathbb{P}(S=0 \mid \mathbf{X}^{\mathrm{m}}=\mathbf{0}, \mathbf{X}^{\mathrm{c}})} \cdot \frac{\mathbb{P}(S=0 \mid \mathbf{X}^{\mathrm{c}})}{\mathbb{P}(S=1 \mid \mathbf{X}^{\mathrm{c}})}}$$

$$= \frac{\mathbb{P}(S = 0 \mid \mathbf{X}^{\mathrm{m}}, \mathbf{X}^{\mathrm{c}}) \cdot \mathbb{P}(S = 1 \mid \mathbf{X}^{\mathrm{c}}) \cdot \mathbb{P}(\mathbf{X}^{\mathrm{m}} \mid \mathbf{X}^{\mathrm{c}}, S = 1)}{\mathbb{P}(S = 1 \mid \mathbf{X}^{\mathrm{m}}, \mathbf{X}^{\mathrm{c}}) \cdot \mathbb{P}(S = 0 \mid \mathbf{X}^{\mathrm{c}})}$$

$$= \frac{\mathbb{P}(S = 0 \mid \mathbf{X}^{\mathrm{m}}, \mathbf{X}^{\mathrm{c}}) \cdot \mathbb{P}(S = 1, \mathbf{X}^{\mathrm{c}}) \cdot \mathbb{P}(\mathbf{X}^{\mathrm{m}} \mid \mathbf{X}^{\mathrm{c}}, S = 1)}{\mathbb{P}(S = 1 \mid \mathbf{X}^{\mathrm{m}}, \mathbf{X}^{\mathrm{c}}) \cdot \mathbb{P}(S = 0, \mathbf{X}^{\mathrm{c}})}$$

$$= \frac{\mathbb{P}(S = 0 \mid \mathbf{X}^{\mathrm{m}}, \mathbf{X}^{\mathrm{c}}) \cdot \mathbb{P}(\mathbf{X}^{\mathrm{m}}, \mathbf{X}^{\mathrm{c}}, S = 1)}{\mathbb{P}(S = 1 \mid \mathbf{X}^{\mathrm{m}}, \mathbf{X}^{\mathrm{c}}) \cdot \mathbb{P}(S = 0, \mathbf{X}^{\mathrm{c}})}$$

$$= \frac{\mathbb{P}(S = 0 \mid \mathbf{X}^{\mathrm{m}}, \mathbf{X}^{\mathrm{c}}) \cdot \mathbb{P}(\mathbf{X}^{\mathrm{m}}, \mathbf{X}^{\mathrm{c}})}{\mathbb{P}(S = 0, \mathbf{X}^{\mathrm{c}})}$$

$$= \mathbb{P}(\mathbf{X}^{\mathrm{m}} \mid \mathbf{X}^{\mathrm{c}}, S = 0).$$

Consequently, the right-hand side of Equation (2) equals the left-hand side, which proves the correctness of Equation (2).

□

# E. Proof of Lemma 3.5

*Proof.* First, substituting Equation (5) into Equation (4) yields

$$
\begin{aligned}
&\mathrm{OR}(\mathbf{X}^{\mathrm{m}}, \mathbf{X}^{\mathrm{c}}) \\
&= \frac{\mathrm{OR}(\mathbf{X}^{\mathrm{m}}, \mathbf{X}^{\mathrm{c}}) \cdot \mathbb{E}[\mathrm{OR}(\mathbf{X}^{\mathrm{m}}, \mathbf{X}^{\mathrm{c}}) \mid \widetilde{\mathbf{X}^{\mathrm{c}}}, S = 1]}{\mathrm{OR}(\mathbf{X}^{\mathrm{m}} = \mathbf{0}, \mathbf{X}^{\mathrm{c}}) \cdot \mathbb{E}[\mathrm{OR}(\mathbf{X}^{\mathrm{m}}, \mathbf{X}^{\mathrm{c}}) \mid \widetilde{\mathbf{X}^{\mathrm{c}}}, S = 1]} \\
&= \frac{\mathrm{OR}(\mathbf{X}^{\mathrm{m}}, \mathbf{X}^{\mathrm{c}})}{\mathrm{OR}(\mathbf{X}^{\mathrm{m}} = \mathbf{0}, \mathbf{X}^{\mathrm{c}})},
\end{aligned}
$$

where $\mathrm{OR}(\mathbf{X}^{\mathrm{m}} = \mathbf{0}, \mathbf{X}^{\mathrm{c}}) = 1$, obtained by substituting $\mathbf{X}^{\mathrm{m}} = \mathbf{0}$ into Equation (3).

Therefore, the right-hand side of Equation (4) equals the left-hand side, which proves the correctness of Equation (4).

Next, under Assumptions 2.5 and 3.2, we have

$$
\mathbb{P}(S \mid \mathbf{X}^{\mathrm{m}}, \mathbf{X}^{\mathrm{c}}) = \mathbb{P}(S \mid \mathbf{X}^{\mathrm{m}}, \widetilde{\mathbf{X}^{\mathrm{c}}}).
$$

Therefore, according to Equation (3), we have

$$
\mathrm{OR}(\mathbf{X}^{\mathrm{m}}, \mathbf{X}^{\mathrm{c}}) = \mathrm{OR}(\mathbf{X}^{\mathrm{m}}, \widetilde{\mathbf{X}^{\mathrm{c}}}).
$$

Based on the proof in Appendix D, we have

$$
\mathbb{E}[\mathrm{OR}(\mathbf{X}^{\mathrm{m}}, \mathbf{X}^{\mathrm{c}}) \mid \widetilde{\mathbf{X}^{\mathrm{c}}}, S = 1] = \frac{\mathbb{P}(S = 1 \mid \mathbf{X}^{\mathrm{m}} = \mathbf{0}, \widetilde{\mathbf{X}^{\mathrm{c}}})}{\mathbb{P}(S = 0 \mid \mathbf{X}^{\mathrm{m}} = \mathbf{0}, \widetilde{\mathbf{X}^{\mathrm{c}}})} \cdot \frac{\mathbb{P}(S = 0 \mid \widetilde{\mathbf{X}^{\mathrm{c}}})}{\mathbb{P}(S = 1 \mid \widetilde{\mathbf{X}^{\mathrm{c}}})}.
$$

Substituting the above three equations into Equation (5) yields

$$
\begin{aligned}
&\widetilde{\mathrm{OR}}(\mathbf{X}^{\mathrm{m}}, \mathbf{X}^{\mathrm{c}}) \\
&= \frac{\mathrm{OR}(\mathbf{X}^{\mathrm{m}}, \widetilde{\mathbf{X}^{\mathrm{c}}})}{\frac{\mathbb{P}(S=1 \mid \mathbf{X}^{\mathrm{m}}=\mathbf{0}, \widetilde{\mathbf{X}^{\mathrm{c}}})}{\mathbb{P}(S=0 \mid \mathbf{X}^{\mathrm{m}}=\mathbf{0}, \widetilde{\mathbf{X}^{\mathrm{c}}})} \cdot \frac{\mathbb{P}(S=0 \mid \widetilde{\mathbf{X}^{\mathrm{c}}})}{\mathbb{P}(S=1 \mid \widetilde{\mathbf{X}^{\mathrm{c}}})}} \\
&= \frac{\mathbb{P}(S = 0 \mid \mathbf{X}^{\mathrm{m}}, \widetilde{\mathbf{X}^{\mathrm{c}}}) \cdot \mathbb{P}(S = 1 \mid \widetilde{\mathbf{X}^{\mathrm{c}}})}{\mathbb{P}(S = 1 \mid \mathbf{X}^{\mathrm{m}}, \widetilde{\mathbf{X}^{\mathrm{c}}}) \cdot \mathbb{P}(S = 0 \mid \widetilde{\mathbf{X}^{\mathrm{c}}})} \\
&= \frac{\mathbb{P}(S = 0 \mid \mathbf{X}^{\mathrm{m}}, \mathbf{X}^{\mathrm{c}}) \cdot \mathbb{P}(S = 1 \mid \widetilde{\mathbf{X}^{\mathrm{c}}})}{\mathbb{P}(S = 1 \mid \mathbf{X}^{\mathrm{m}}, \mathbf{X}^{\mathrm{c}}) \cdot \mathbb{P}(S = 0 \mid \widetilde{\mathbf{X}^{\mathrm{c}}})}.
\end{aligned}
$$

Therefore, under Assumptions 2.5 and 3.2, we have

$$
\begin{aligned}
&\mathbb{E}[\widetilde{\mathrm{OR}}(\mathbf{X}^{\mathrm{m}}, \mathbf{X}^{\mathrm{c}}) \mid \mathbf{X}^{\mathrm{c}}, S = 1] \\
&= \mathbb{E}\left[\left.\frac{\mathbb{P}(S = 0 \mid \mathbf{X}^{\mathrm{m}}, \mathbf{X}^{\mathrm{c}}) \cdot \mathbb{P}(S = 1 \mid \widetilde{\mathbf{X}^{\mathrm{c}}})}{\mathbb{P}(S = 1 \mid \mathbf{X}^{\mathrm{m}}, \mathbf{X}^{\mathrm{c}}) \cdot \mathbb{P}(S = 0 \mid \widetilde{\mathbf{X}^{\mathrm{c}}})}\right| \mathbf{X}^{\mathrm{c}}, S = 1\right] \\
&= \mathbb{E}\left[\left.\frac{\mathbb{P}(S = 0 \mid \mathbf{X}^{\mathrm{m}}, \mathbf{X}^{\mathrm{c}}) \cdot \mathbb{P}(S = 1 \mid \widetilde{\mathbf{X}^{\mathrm{c}}})}{\mathbb{P}(S = 1 \mid \mathbf{X}^{\mathrm{m}}, \mathbf{X}^{\mathrm{c}}) \cdot \mathbb{P}(S = 0 \mid \widetilde{\mathbf{X}^{\mathrm{c}}})}\right| \mathbf{X}^{\mathrm{c}}\right] \\
&= \sum_{\mathbf{x}^{\mathrm{m}}} \frac{\mathbb{P}(S = 0 \mid \mathbf{X}^{\mathrm{m}} = \mathbf{x}^{\mathrm{m}}, \mathbf{X}^{\mathrm{c}}) \cdot \mathbb{P}(S = 1 \mid \widetilde{\mathbf{X}^{\mathrm{c}}})}{\mathbb{P}(S = 1 \mid \mathbf{X}^{\mathrm{m}} = \mathbf{x}^{\mathrm{m}}, \mathbf{X}^{\mathrm{c}}) \cdot \mathbb{P}(S = 0 \mid \widetilde{\mathbf{X}^{\mathrm{c}}})} \cdot \mathbb{P}(\mathbf{X}^{\mathrm{m}} = \mathbf{x}^{\mathrm{m}} \mid \mathbf{X}^{\mathrm{c}}) \\
&= \frac{\mathbb{P}(S = 1 \mid \widetilde{\mathbf{X}^{\mathrm{c}}})}{\mathbb{P}(S = 0 \mid \widetilde{\mathbf{X}^{\mathrm{c}}})} \cdot \sum_{\mathbf{x}^{\mathrm{m}}} \frac{\mathbb{P}(S = 0 \mid \mathbf{X}^{\mathrm{m}} = \mathbf{x}^{\mathrm{m}}, \mathbf{X}^{\mathrm{c}})}{\mathbb{P}(S = 1 \mid \mathbf{X}^{\mathrm{m}} = \mathbf{x}^{\mathrm{m}}, \mathbf{X}^{\mathrm{c}})} \cdot \mathbb{P}(\mathbf{X}^{\mathrm{m}} = \mathbf{x}^{\mathrm{m}} \mid \mathbf{X}^{\mathrm{c}}) \\
&= \frac{\mathbb{P}(S = 1 \mid \widetilde{\mathbf{X}^{\mathrm{c}}}) \cdot \mathbb{P}(S = 0 \mid \mathbf{X}^{\mathrm{c}})}{\mathbb{P}(S = 0 \mid \widetilde{\mathbf{X}^{\mathrm{c}}}) \cdot \mathbb{P}(S = 1 \mid \mathbf{X}^{\mathrm{c}})} \\
&= \frac{\mathbb{P}(S = 1, \widetilde{\mathbf{X}^{\mathrm{c}}}) \cdot \mathbb{P}(S = 0, \widetilde{\mathbf{X}^{\mathrm{c}}}, \mathbf{Z})}{\mathbb{P}(S = 0, \widetilde{\mathbf{X}^{\mathrm{c}}}) \cdot \mathbb{P}(S = 1, \widetilde{\mathbf{X}^{\mathrm{c}}}, \mathbf{Z})} \\
&= \frac{\mathbb{P}(\mathbf{Z} \mid \widetilde{\mathbf{X}^{\mathrm{c}}}, S = 0)}{\mathbb{P}(\mathbf{Z} \mid \widetilde{\mathbf{X}^{\mathrm{c}}}, S = 1)}.
\end{aligned}
$$

Consequently, the right-hand side of Equation (6) equals the left-hand side, which proves the correctness of Equation (6).

Equation (6) is a Fredholm integral equation of the first kind, with $\widetilde{\mathrm{OR}}(\mathbf{X}^{\mathrm{m}}, \mathbf{X}^{\mathrm{c}})$ to be solved for. Based on Theorem 1 from Miao et al. (2024) and Lemma 3.4, Equation (6) has a unique solution under Assumptions 2.5 and 3.2. Therefore, $\mathrm{OR}(\mathbf{X}^{\mathrm{m}}, \mathbf{X}^{\mathrm{c}})$ is identifiable. $\qquad\square$

## F. Proof of Corollary 3.7

*Proof.* From Equation (3), it follows that

$$
\mathbb{P}(S = 1 \mid \mathbf{X}) = \left(1 + \mathrm{OR}(\mathbf{X}^{\mathrm{m}}, \mathbf{X}^{\mathrm{c}}) \cdot \frac{\mathbb{P}(S = 0 \mid \mathbf{X}^{\mathrm{m}} = \mathbf{0}, \mathbf{X}^{\mathrm{c}})}{\mathbb{P}(S = 1 \mid \mathbf{X}^{\mathrm{m}} = \mathbf{0}, \mathbf{X}^{\mathrm{c}})}\right)^{-1}.
$$

The identifiability of this equation depends on the identifiability of $\mathrm{OR}(\mathbf{X}^{\mathrm{m}}, \mathbf{X}^{\mathrm{c}})$ and $\mathbb{P}(S = 1 \mid \mathbf{X}^{\mathrm{m}} = \mathbf{0}, \mathbf{X}^{\mathrm{c}})$, both of which can be identified with $\mathbf{Z}$ under Assumptions 2.3, 2.4, 2.5, 2.6, 3.1, and 3.2.

First, under Assumptions 2.5 and 3.2, we have

$$
\mathbb{P}(S = 1 \mid \mathbf{X}^{\mathrm{m}} = \mathbf{0}, \mathbf{X}^{\mathrm{c}}) = \left(1 + \frac{\mathbb{P}(S = 0 \mid \widetilde{\mathbf{X}^{\mathrm{c}}})}{\mathbb{P}(S = 1 \mid \widetilde{\mathbf{X}^{\mathrm{c}}}) \cdot \frac{\mathrm{OR}(\mathbf{X}^{\mathrm{m}}, \mathbf{X}^{\mathrm{c}})}{\widetilde{\mathrm{OR}}(\mathbf{X}^{\mathrm{m}}, \mathbf{X}^{\mathrm{c}})}}\right)^{-1}.
$$

The correctness of the above equation is guaranteed by substituting Equations (3) and (5) and the corresponding equation of $\mathbb{E}[\mathrm{OR}(\mathbf{X}^{\mathrm{m}}, \mathbf{X}^{\mathrm{c}}) \mid \widetilde{\mathbf{X}^{\mathrm{c}}}, S = 1]$ in Appendix E into its right-hand side, which equals the left-hand side.

Next, based on Lemma 3.5, $\mathrm{OR}(\mathbf{X}^{\mathrm{m}}, \mathbf{X}^{\mathrm{c}})$ and $\widetilde{\mathrm{OR}}(\mathbf{X}^{\mathrm{m}}, \mathbf{X}^{\mathrm{c}})$ are identifiable under Assumptions 2.3, 2.4, 2.5, 2.6, 3.1, and 3.2, which consequently ensures the identifiability of $\mathbb{P}(S = 1 \mid \mathbf{X}^{\mathrm{m}} = \mathbf{0}, \mathbf{X}^{\mathrm{c}})$.

With both $\mathbb{P}(S = 1 \mid \mathbf{X}^{\mathrm{m}} = \mathbf{0}, \mathbf{X}^{\mathrm{c}})$ and $\mathrm{OR}(\mathbf{X}^{\mathrm{m}}, \mathbf{X}^{\mathrm{c}})$ identified, $\mathbb{P}(S = 1 \mid \mathbf{X})$ is identified as

$$
\mathbb{P}(S = 1 \mid \mathbf{X}) = \left( 1 + \mathrm{OR}(\mathbf{X}^{\mathrm{m}}, \mathbf{X}^{\mathrm{c}}) \cdot \frac{\frac{\mathbb{P}(S=0|\widetilde{\mathbf{X}^{\mathrm{c}}})}{\mathbb{P}(S=1|\widetilde{\mathbf{X}^{\mathrm{c}}}) \cdot \frac{\mathrm{OR}(\mathbf{X}^{\mathrm{m}},\mathbf{X}^{\mathrm{c}})}{\widetilde{\mathrm{OR}}(\mathbf{X}^{\mathrm{m}},\mathbf{X}^{\mathrm{c}})} + \mathbb{P}(S=0|\widetilde{\mathbf{X}^{\mathrm{c}}})}}{\frac{\mathbb{P}(S=1|\widetilde{\mathbf{X}^{\mathrm{c}}}) \cdot \frac{\mathrm{OR}(\mathbf{X}^{\mathrm{m}},\mathbf{X}^{\mathrm{c}})}{\widetilde{\mathrm{OR}}(\mathbf{X}^{\mathrm{m}},\mathbf{X}^{\mathrm{c}})}}{\mathbb{P}(S=1|\widetilde{\mathbf{X}^{\mathrm{c}}}) \cdot \frac{\mathrm{OR}(\mathbf{X}^{\mathrm{m}},\mathbf{X}^{\mathrm{c}})}{\widetilde{\mathrm{OR}}(\mathbf{X}^{\mathrm{m}},\mathbf{X}^{\mathrm{c}})} + \mathbb{P}(S=0|\widetilde{\mathbf{X}^{\mathrm{c}}})}} \right)^{-1}
$$

$$
= \left( 1 + \mathrm{OR}(\mathbf{X}^{\mathrm{m}}, \mathbf{X}^{\mathrm{c}}) \cdot \frac{\mathbb{P}(S = 0 \mid \widetilde{\mathbf{X}^{\mathrm{c}}})}{\mathbb{P}(S = 1 \mid \widetilde{\mathbf{X}^{\mathrm{c}}}) \cdot \frac{\mathrm{OR}(\mathbf{X}^{\mathrm{m}},\mathbf{X}^{\mathrm{c}})}{\widetilde{\mathrm{OR}}(\mathbf{X}^{\mathrm{m}},\mathbf{X}^{\mathrm{c}})}} \right)^{-1}
$$

$$
= \left( 1 + \widetilde{\mathrm{OR}}(\mathbf{X}^{\mathrm{m}}, \mathbf{X}^{\mathrm{c}}) \cdot \frac{\mathbb{P}(S = 0 \mid \widetilde{\mathbf{X}^{\mathrm{c}}})}{\mathbb{P}(S = 1 \mid \widetilde{\mathbf{X}^{\mathrm{c}}})} \right)^{-1}
$$

Moreover, $\mathbb{P}(R = 1 \mid \mathbf{X})$ is identified as

$$
\mathbb{P}(R = 1 \mid \mathbf{X}) = \begin{cases} \mathbb{P}(S = 1 \mid \mathbf{X}) & \text{for Setting 1} \\ 1 - \mathbb{P}(S = 1 \mid \mathbf{X}) & \text{for Setting 2} \end{cases}
$$

since $R = S$ in Setting 1, and $R = 1 - S$ in Setting 2. $\qquad \square$

## G. Proof of Theorem 4.2

*Proof.* We only need to prove the unbiasedness of $\hat{\psi}$, and then the consistency of $\hat{\psi}$ in large samples holds under the conditions specified in Newey & McFadden (1994); Miao & Tchetgen Tchetgen (2016).

**Step 1. Unbiasedness of the odds ratio model.**

Given the conditions specified in Silverman (2018), $\hat{f}(\mathbf{Z} \mid \widetilde{\mathbf{X}^{\mathrm{c}}}, S)$ is a consistent estimate of $\mathbb{P}(\mathbf{Z} \mid \widetilde{\mathbf{X}^{\mathrm{c}}}, S)$.

Therefore, based on Equation (6), if the $\widetilde{\mathrm{OR}}$ model is correctly specified, the following equation holds.

$$
\mathbb{E}\left[ \widetilde{\mathrm{OR}}(\mathbf{X}^{\mathrm{m}}, \mathbf{X}^{\mathrm{c}}) - \frac{\hat{f}(\mathbf{Z} \mid \widetilde{\mathbf{X}^{\mathrm{c}}}, S = 0)}{\hat{f}(\mathbf{Z} \mid \widetilde{\mathbf{X}^{\mathrm{c}}}, S = 1)} \right] = 0.
$$

Therefore, the unbiasedness of the $\widetilde{\mathrm{OR}}$ model holds, and thus the unbiasedness of the odds ratio model also holds based on Equation (4).

**Step 2. Unbiasedness of the doubly robust imputation model.**

To prove the unbiasedness of $\hat{\psi}(\mathbf{X}^{\mathrm{c}})$, we need to prove that $\mathbb{E}\left[ \hat{\psi}(\mathbf{X}^{\mathrm{c}}) - \mathbf{X}^{\mathrm{m}} \right] = 0$ holds.

Under Assumption 3.2, it reduces to proving that the following equation holds.

$$
\begin{aligned}
\mathbb{E}\left[ \hat{\psi}(\mathbf{X}^{\mathrm{c}}) - \mathbf{X}^{\mathrm{m}} \right] &= \mathbb{E}\left[ \left( S \cdot \left( \frac{\mathbf{X}^{\mathrm{m}} - \hat{\delta}(\mathbf{X}^{\mathrm{c}})}{\hat{\pi}_s(\mathbf{X})} \right) + \hat{\delta}(\mathbf{X}^{\mathrm{c}}) \right) - \mathbf{X}^{\mathrm{m}} \right] \\
&= \mathbb{E}\left[ \left( \frac{S}{\hat{\pi}_s(\mathbf{X})} - 1 \right) \cdot \left( \mathbf{X}^{\mathrm{m}} - \hat{\delta}(\mathbf{X}^{\mathrm{c}}) \right) \right] \\
&= \mathbb{E}\left[ \mathbb{E}\left[ \left( \frac{S}{\hat{\pi}_s(\mathbf{X})} - 1 \right) \cdot \left( \mathbf{X}^{\mathrm{m}} - \hat{\delta}(\mathbf{X}^{\mathrm{c}}) \right) \Big| \mathbf{X}^{\mathrm{c}}, \mathbf{X}^{\mathrm{m}} \right] \right] \\
&= \mathbb{E}\left[ \left( \frac{\mathbb{P}(S = 1 \mid \mathbf{X})}{\hat{\pi}_s(\mathbf{X})} - 1 \right) \cdot \left( \mathbf{X}^{\mathrm{m}} - \hat{\delta}(\mathbf{X}^{\mathrm{c}}) \right) \right] \\
&= 0.
\end{aligned}
$$

We provide proofs for the following two cases.

**(1) The regression model of the missing covariates is correctly specified while the selection score model is not.** Based on Equation (2), we have

$$
\mathbb{E}[\mathbf{X}^{\mathrm{m}} \mid \mathbf{X}^{\mathrm{c}}, S = 0] = \sum_{\mathbf{x}^{\mathrm{m}}} \mathbf{x}^{\mathrm{m}} \cdot \mathbb{P}(\mathbf{X}^{\mathrm{m}} = \mathbf{x}^{\mathrm{m}} \mid \mathbf{X}^{\mathrm{c}}, S = 0)
$$

$$
= \sum_{\mathbf{x}^{\mathrm{m}}} \mathbf{x}^{\mathrm{m}} \cdot \frac{\mathrm{OR}(\mathbf{X}^{\mathrm{m}} = \mathbf{x}^{\mathrm{m}}, \mathbf{X}^{\mathrm{c}}) \cdot \mathbb{P}(\mathbf{X}^{\mathrm{m}} = \mathbf{x}^{\mathrm{m}} \mid \mathbf{X}^{\mathrm{c}}, S = 1)}{\mathbb{E}[\mathrm{OR}(\mathbf{X}^{\mathrm{m}}, \mathbf{X}^{\mathrm{c}}) \mid \mathbf{X}^{\mathrm{c}}, S = 1]}
$$

$$
= \sum_{\mathbf{x}^{\mathrm{m}}} \mathbf{x}^{\mathrm{m}} \cdot \frac{\mathrm{OR}(\mathbf{X}^{\mathrm{m}} = \mathbf{x}^{\mathrm{m}}, \mathbf{X}^{\mathrm{c}}) \cdot \mathbb{P}(\mathbf{X}^{\mathrm{m}} = \mathbf{x}^{\mathrm{m}} \mid \mathbf{X}^{\mathrm{c}}) \cdot \mathbb{P}(S = 1 \mid \mathbf{X}^{\mathrm{m}} = \mathbf{x}^{\mathrm{m}}, \mathbf{X}^{\mathrm{c}})}{\mathbb{E}[\mathrm{OR}(\mathbf{X}^{\mathrm{m}}, \mathbf{X}^{\mathrm{c}}) \mid \mathbf{X}^{\mathrm{c}}, S = 1] \cdot \mathbb{P}(S = 1 \mid \mathbf{X}^{\mathrm{c}})}
$$

$$
= \frac{\mathbb{E}\left[(S \cdot \mathrm{OR}(\mathbf{X}^{\mathrm{m}}, \mathbf{X}^{\mathrm{c}}) \cdot \mathbf{X}^{\mathrm{m}}) \mid \mathbf{X}^{\mathrm{c}}\right]}{\mathbb{E}[S \cdot \mathrm{OR}(\mathbf{X}^{\mathrm{m}}, \mathbf{X}^{\mathrm{c}}) \mid \mathbf{X}^{\mathrm{c}}]}.
$$

Therefore, we have

$$
\mathbb{E}\left[(S \cdot \mathrm{OR}(\mathbf{X}^{\mathrm{m}}, \mathbf{X}^{\mathrm{c}}) \cdot \mathbf{X}^{\mathrm{m}}) \mid \mathbf{X}^{\mathrm{c}}\right] = \mathbb{E}[\mathbf{X}^{\mathrm{m}} \mid \mathbf{X}^{\mathrm{c}}, S = 0] \cdot \mathbb{E}[S \cdot \mathrm{OR}(\mathbf{X}^{\mathrm{m}}, \mathbf{X}^{\mathrm{c}}) \mid \mathbf{X}^{\mathrm{c}}]
$$

and

$$
\mathbb{E}\left[(S \cdot \mathrm{OR}(\mathbf{X}^{\mathrm{m}}, \mathbf{X}^{\mathrm{c}}) \cdot (\mathbf{X}^{\mathrm{m}} - \mathbb{E}[\mathbf{X}^{\mathrm{m}} \mid \mathbf{X}^{\mathrm{c}}, S = 0])) \mid \mathbf{X}^{\mathrm{c}}\right]
$$

$$
= \mathbb{E}\left[(S \cdot \mathrm{OR}(\mathbf{X}^{\mathrm{m}}, \mathbf{X}^{\mathrm{c}}) \cdot \mathbf{X}^{\mathrm{m}}) \mid \mathbf{X}^{\mathrm{c}}\right] - \mathbb{E}\left[(S \cdot \mathrm{OR}(\mathbf{X}^{\mathrm{m}}, \mathbf{X}^{\mathrm{c}}) \cdot \mathbb{E}[\mathbf{X}^{\mathrm{m}} \mid \mathbf{X}^{\mathrm{c}}, S = 0]) \mid \mathbf{X}^{\mathrm{c}}\right]
$$

$$
= \mathbb{E}\left[(S \cdot \mathrm{OR}(\mathbf{X}^{\mathrm{m}}, \mathbf{X}^{\mathrm{c}}) \cdot \mathbf{X}^{\mathrm{m}}) \mid \mathbf{X}^{\mathrm{c}}\right] - \mathbb{E}[\mathbf{X}^{\mathrm{m}} \mid \mathbf{X}^{\mathrm{c}}, S = 0] \cdot \mathbb{E}[S \cdot \mathrm{OR}(\mathbf{X}^{\mathrm{m}}, \mathbf{X}^{\mathrm{c}}) \mid \mathbf{X}^{\mathrm{c}}]
$$

$$
= 0
$$

Then, we have

$$
\mathbb{E}\left[\left(\frac{S}{\hat{\pi}_s(\mathbf{X})} - 1\right) \cdot (\mathbf{X}^{\mathrm{m}} - \mathbb{E}[\mathbf{X}^{\mathrm{m}} \mid \mathbf{X}^{\mathrm{c}}, S = 0])\right]
$$

$$
= \mathbb{E}\left[\left(S \cdot \mathrm{OR}(\mathbf{X}^{\mathrm{m}}, \mathbf{X}^{\mathrm{c}}) \cdot \frac{\mathbb{P}(S = 0 \mid \mathbf{X}^{\mathrm{m}} = \mathbf{0}, \mathbf{X}^{\mathrm{c}})}{\mathbb{P}(S = 1 \mid \mathbf{X}^{\mathrm{m}} = \mathbf{0}, \mathbf{X}^{\mathrm{c}})} \cdot (\mathbf{X}^{\mathrm{m}} - \mathbb{E}[\mathbf{X}^{\mathrm{m}} \mid \mathbf{X}^{\mathrm{c}}, S = 0])\right)\right]
$$

$$
= \mathbb{E}\left[\mathbb{E}\left[\left(S \cdot \mathrm{OR}(\mathbf{X}^{\mathrm{m}}, \mathbf{X}^{\mathrm{c}}) \cdot \frac{\mathbb{P}(S = 0 \mid \mathbf{X}^{\mathrm{m}} = \mathbf{0}, \mathbf{X}^{\mathrm{c}})}{\mathbb{P}(S = 1 \mid \mathbf{X}^{\mathrm{m}} = \mathbf{0}, \mathbf{X}^{\mathrm{c}})} \cdot (\mathbf{X}^{\mathrm{m}} - \mathbb{E}[\mathbf{X}^{\mathrm{m}} \mid \mathbf{X}^{\mathrm{c}}, S = 0])\right) \middle| \mathbf{X}^{\mathrm{c}}\right]\right]
$$

$$
= 0
$$

Given the above equations, as the regression model of $\mathbf{X}^{\mathrm{m}}$ is correctly specified and the odds ratio model is unbiasedly estimated, $\hat{\delta}(\mathbf{X}^{\mathrm{c}})$ is an unbiased estimate of $\mathbb{E}[\mathbf{X}^{\mathrm{m}} \mid \mathbf{X}^{\mathrm{c}}, S = 0]$. Consequently, we have

$$
\mathbb{E}\left[\left(\frac{S}{\hat{\pi}_s(\mathbf{X})} - 1\right) \cdot \left(\mathbf{X}^{\mathrm{m}} - \hat{\delta}(\mathbf{X}^{\mathrm{c}})\right)\right] = 0.
$$

Therefore, the unbiasedness of $\hat{\psi}(\mathbf{X}^{\mathrm{c}})$ holds.

**(2) The selection score model is correctly specified while the regression model of the missing covariates is not.** In this case, as the selection score model of $S$ is correctly specified, $\hat{\gamma}(\mathbf{X})$ is an unbiased estimate of $\mathbb{P}(S = 1 \mid \widetilde{\mathbf{X}^{\mathrm{c}}})$. Therefore, given the unbiasedness of the odds ratio model, based on Corollary 3.7, the unbiasedness of $\hat{\pi}_s(\mathbf{X})$ holds under Assumptions 2.3, 2.4, 2.5, 2.6, 3.1, and 3.2. Consequently, we have

$$
\frac{\mathbb{P}(S = 1 \mid \mathbf{X})}{\hat{\pi}_s(\mathbf{X})} - 1 = 0.
$$

Therefore, the unbiasedness of $\hat{\psi}(\mathbf{X}^{\mathrm{c}})$ holds. □

## H. Proof of Theorem 4.3

*Proof.* When either $\hat{\pi}_r(\mathbf{X})$ or $\hat{\mu}_t(\mathbf{X})$ is consistent, the consistency theory of the doubly robust TATE estimator in large samples has already been established in Little & Rubin (2019). Therefore, we focus on proving the consistency of $\hat{\pi}_r(\mathbf{X})$ and $\hat{\mu}_t(\mathbf{X})$ here.

(1) Under Assumptions 2.3, 2.4, 2.5, 2.6, 3.1, and 3.2, if the selection score model of $S$ and the $\widetilde{\mathrm{OR}}$ model are correctly specified, $\hat{\pi}_s(\mathbf{X})$ is consistent, as proved in Appendix G. Since $\hat{\pi}_r(\mathbf{X})$ equals $\hat{\pi}_s(\mathbf{X})$ or $1 - \hat{\pi}_s(\mathbf{X})$ based on Corollary 3.7, $\hat{\pi}_r(\mathbf{X})$ is also consistent.

(2) Given that the imputation model is consistent, if the outcome regression model is correctly specified, $\hat{\mu}_t(\mathbf{X})$ estimated using the imputed data still maintains consistency under the conditions specified in Newey & McFadden (1994); Little & Rubin (2019).

Therefore, at least one of $\hat{\pi}_r(\mathbf{X})$ and $\hat{\mu}_t(\mathbf{X})$ is consistent, and thus $\hat{\tau}$ is consistent. $\qquad \square$

## I. Time Complexity Analysis of the Shadow Variable Selection Process in Stage I of 2SDR

In Setting 1, where the values of $\mathbf{X}^m$ are missing in Dataset $\mathcal{O}$, the time complexity of the initial screening procedure is $O(n_\mathcal{R} \cdot k^2)$ (Zou, 2006), where $k = p - d$ is the number of variables in $\mathbf{X}^c$, and $p$ is the number of variables in $\mathbf{X}$; the time complexity of the test for Assumption3.2(1) is $O(n_\mathcal{O} \cdot p^2 + n_\mathcal{R} \cdot p^2)$ (Strobl et al., 2019); and the time complexity of the test for Assumption3.2(2), which uses the mini-batch stochastic gradient descent method for optimization, is $O(n_\mathcal{R} \cdot k^2)$ (Bottou et al., 1991). Similarly, in Setting 2, where the values of $\mathbf{X}^m$ are missing in Dataset $\mathcal{R}$, the time complexity of the initial screening procedure is $O(n_\mathcal{O} \cdot k^2)$; the time complexity of the test for Assumption3.2(1) is $O(mp^2 + np^2)$; and the time complexity of the test for Assumption3.2(1) is $O(n_\mathcal{O} \cdot p^2 + n_\mathcal{R} \cdot p^2)$. As a result, the time complexity of the entire shadow variable selection process is $O(np^2) \equiv O(n_\mathcal{O} \cdot p^2 + n_\mathcal{R} \cdot p^2)$.

The time complexity may vary depending on the chosen method for testing conditional independence. Our approach is flexible and can incorporate any Conditional Independence Testing (CIT) method, with the specific choice of method depending on its characteristics and the practical context (Li & Fan, 2020; Zheng et al., 2024).

## J. Experimental Details

### J.1. Software and Hardware Used

**Software used:** Python 3.9 with PyTorch 1.13.0.

**Hardware used:** Windows 11 operating system with a 13th Gen Intel(R) Core(TM) i7-13700K CPU and an NVIDIA GeForce RTX 3080 GPU (with CUDA version 12.1).

### J.2. Implementation Details

In the experiments conducted on the low-dimensional synthetic datasets and the real-world datasets, we used Elastic Nets (Zou & Hastie, 2005) for continuous variable estimation and logistic regression for binary variable estimation. In the experiments conducted on the high-dimensional synthetic dataset, neural networks were employed for these tasks, using the Adam optimizer (Kingma & Ba, 2015) with the initial learning rate being 0.003. We used cross-validation to determine the hyperparameters. Based on Theorem 3.3, for the methods involving imputation, during the cross-validation process on the datasets under Setting 2, we evaluated the performance of the models trained on the RCT training set not only on the RCT validation set but also on the validation set from the observational data from the target population to ensure the accuracy of the imputation.

## K. Data Generation Process for the Synthetic Datasets

### K.1. Data Generation Process for the Low-Dimensional Datasets

We first used a simulation framework similar to that in Colnet et al. (2022) to generate the covariates. Specifically, we generated $X_2 \sim \mathcal{N}(0, 1)$ and generated $X_1$, $X_3$, $X_4$, and $X_5$ as

$$X_1, X_4 \sim \mathcal{N}\left(\begin{pmatrix} 0 \\ 0 \end{pmatrix}, \begin{pmatrix} 1 & 0.8 \\ 0.8 & 1 \end{pmatrix}\right)$$

and

$$X_3, X_5 \sim \mathcal{N}\left(\begin{pmatrix} 0 \\ 0 \end{pmatrix}, \begin{pmatrix} 1 & 0.8 \\ 0.8 & 1 \end{pmatrix}\right).$$

Table 3. TATE estimation results (MAE) on the high-dimensional synthetic dataset (mean±std), with bold values indicating the best performance.

| METHODS | SETTING 1 | SETTING 2 |
|---|---|---|
| RCT | 5.049±2.166 | 5.623±2.284 |
| IPSW | 8.530±3.741 | 9.412±4.631 |
| CW | 74.47±48.95 | 66.51±31.88 |
| G-FORMULA | 0.670±0.365 | 0.686±0.462 |
| AIPSW | 0.664±0.367 | 0.686±0.476 |
| ACW | 0.668±0.366 | 0.685±0.460 |
| IPSW$_{\text{Imp}}$ | 8.173±3.752 | 9.533±3.296 |
| CW$_{\text{Imp}}$ | 79.23±64.57 | 9.859±6.671 |
| G-FORMULA$_{\text{Imp}}$ | 0.612±0.366 | 1.518±0.750 |
| AIPSW$_{\text{Imp}}$ | 0.598±0.376 | 4.542±3.860 |
| ACW$_{\text{Imp}}$ | 0.612±0.365 | 1.518±0.751 |
| 2SDR | **0.433±0.291** | **0.461±0.276** |

Next, to simulate environmental shifts between the RCT data and the observational data from the target population, we generated $R$ using the following logit model:

$$\text{logit}(\mathbb{P}(R = 1 \mid X_1, X_3, X_4)) = -0.5X_1 + 0.5X_3 - 0.3X_4,$$

where $\epsilon_s \sim \mathcal{N}(0, 1)$.

In Setting 1, the set of observable covariates in the RCT data is $\{X_1, X_3, X_4, X_5\}$, while the set of observable covariates in the observational data from the target population is $\{X_1, X_4, X_5\}$, with $X_2$ being unobservable in both datasets. In Setting 2, the set of observable covariates in the RCT datas is $\{X_1, X_4, X_5\}$, while the set of observable covariates in the observational data is $\{X_1, X_3, X_4, X_5\}$.

For the RCT data, we randomly generated $T \sim \mathcal{B}(0.5)$, where $\mathcal{B}(\cdot)$ denotes the Bernoulli distribution. In contrast, for the observational data from the target population, all values of $T$ were set to 0.

Finally, we generated $Y$ as

$$Y = \sum_{i=1}^{5} 5X_i + T \cdot (5X_1 + 5X_2 - 3X_3) + \epsilon_y,$$

where $\epsilon_y \sim \mathcal{N}(0, 1)$.

### K.2. Data Generation Process for the High-dimensional Dataset

The generation of $R$ and $Y$ in the high-dimensional dataset followed the same process as in the low-dimensional datasets, with a slight modification in the generation of the covariates. Specifically, we first generated $\mathbf{X}_1 \sim \mathcal{N}(\mathbf{0}, \mathbf{I}_{20})$, $\mathbf{X}_2 \sim \mathcal{N}(\mathbf{0}, \mathbf{I}_5)$, and $\mathbf{X}_5 \sim \mathcal{N}(\mathbf{0}, \mathbf{I}_{10})$, where $\mathbf{I}_e$ denotes the identity matrix with $e$ rows and $e$ columns. Subsequently, we generated $\mathbf{X}_3$ and $\mathbf{X}_4$ as

$$\mathbf{X}_3 = \mathbf{X}_5 \mathbf{A},$$
$$\mathbf{X}_4 = \mathbf{X}_1 \mathbf{B},$$

where $\mathbf{A}$ is a $10 \times 5$ matrix with each element sampled from $\mathcal{N}(0, 1)$, and $\mathbf{B}$ is a $20 \times 10$ matrix with each element sampled from $\mathcal{N}(0, 1)$.

In the high-dimensional dataset, we set $n_{\mathcal{R}} = 2000$ and $n_{\mathcal{O}} = 10000$. The final dataset contains 50 covariates, including 5 partially observable, 5 completely unobservable, and 40 fully observable variables.

*Table 4.* TATE estimation results (MAE) of 2SDR under different degrees of violation of Assumption 3.2 (mean±std).

| DEGREES | CASE 1 | CASE 2 |
|---------|--------|--------|
| SEVERE | 0.355±0.319 | 0.383±0.280 |
| SLIGHT | 0.311±0.293 | 0.341±0.281 |
| NONE | 0.268±0.212 | 0.268±0.212 |

*Table 5.* Imputation error (MSE) on different datasets (mean±std), with bold values indicating the better performance.

| DATASETS | BASELINE | 2SDR |
|----------|----------|------|
| LOW-DIM$_{2000}$ SETTING 1 | 0.684±0.027 | **0.509±0.008** |
| LOW-DIM$_{1000}$ SETTING 1 | 0.694±0.044 | **0.552±0.038** |
| LOW-DIM$_{500}$ SETTING 1 | 0.786±0.060 | **0.555±0.037** |
| LOW-DIM$_{2000}$ SETTING 2 | 0.386±0.013 | **0.230±0.009** |
| LOW-DIM$_{1000}$ SETTING 2 | 0.397±0.019 | **0.245±0.011** |
| LOW-DIM$_{500}$ SETTING 2 | 0.447±0.023 | **0.310±0.057** |
| HIGH-DIM SETTING 1 | 0.900±0.020 | **0.418±0.186** |
| HIGH-DIM SETTING 2 | 0.986±0.034 | **0.651±0.044** |
| ACTG | 1.897±0.180 | **0.454±0.032** |
| JTPA | 2.130±0.111 | **0.992±0.032** |

# L. Supplementary Experimental Results

## L.1. Experimental Results on the High-Dimensional Synthetic Dataset

We report the TATE estimation results on the high-dimensional synthetic dataset in Table 1. It can be observed that, although, like all the baselines, the performance of 2SDR gets worse compared to the results on the corresponding low-dimensional dataset, it still achieves the best performance, demonstrating its effectiveness in handling high-dimensional data.

## L.2. Robustness Analysis when Assumption 3.2 Is Violated

To evaluate the robustness of 2SDR when the core assumption required by it—Assumption 3.2—is violated, we modified the synthetic datasets to no longer satisfy Assumption 3.2 and conducted experiments on them to observe the performance of 2SDR. Specifically, we modified the synthetic datasets to simulate the following cases:

- **Case 1: Assumption 3.2(2) is violated.** We changed the coefficient of $X_5$ on $R$ from 0 (indicating no violation of the assumption, represented by 'None') to $\{0.1$ (indicating a slight violation of the assumption, represented by 'Slight'), $0.3$ (indicating a severe violation of the assumption, represented by 'Severe')$\}$ to introduce weak dependence between the shadow variable and environmental shifts.
- **Case 2: Assumption 3.2(2) is violated.** We reduced the correlation coefficient between $X_5$ and $X_3$ from $0.8$ (None) to $\{0.3$ (Slight), $0.1$ (Severe)$\}$, so that the shadow variable only has a weak predictive ability for the missing covariate.

As shown in Table 4, while the performance of 2SDR deteriorates when Assumption 3.2 is violated, it does so progressively as the degree of violation increases. The results demonstrate that 2SDR maintains a certain level of robustness even when Assumption 3.2 is not satisfied.

## L.3. Imputation Accuracy Analysis

To evaluate the imputation accuracy of 2SDR, we compare the imputation error (Mean-Square Error, MSE) of 2SDR with that of the baselines performing imputation based on $\mathbb{E}[\mathbf{X}^m \mid \mathbf{X}^c, S = 1]$, as reported in Table 5. The results show that an increase in the dimensionality of the data and a decrease in the RCT scale lead to an increase in the imputation error. However, 2SDR consistently outperforms the baseline on all datasets, demonstrating its superior imputation accuracy.

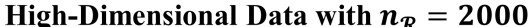

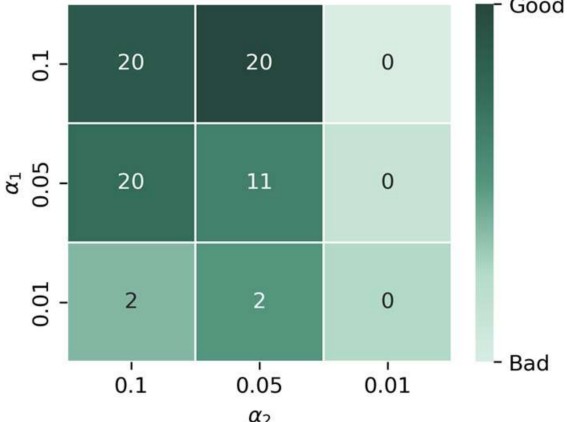

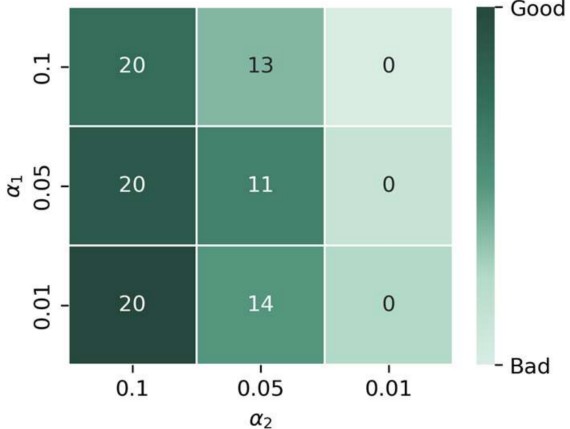

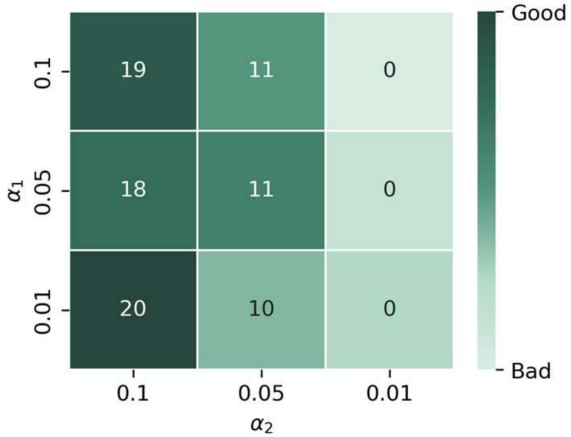

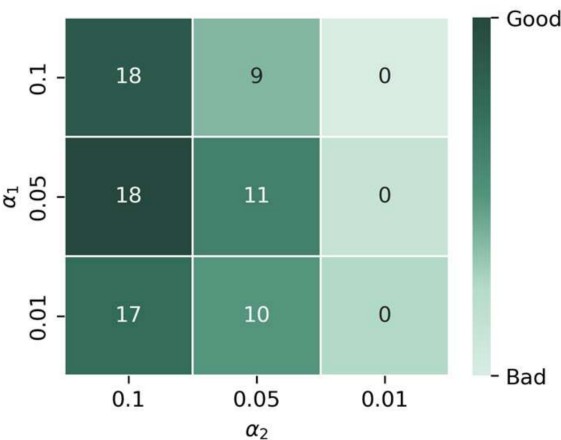

*Figure 4.* The number of times the shadow variables selected by 2SDR were correct within the 20 repetitions of each experiment on different synthetic datasets under Setting 1 varied with different thresholds.

## L.4. Hyperparameter Analysis

In the shadow variable selection process of the first stage of 2SDR, we introduce two hyperparameters, $\alpha_1$ and $\alpha_2$, as the reject thresholds in hypothesis testing. In the experiments on the synthetic datasets under Setting 1, we varied their values to investigate the impact of different thresholds on the accuracy of the selected shadow variables, considering different sample sizes and dimensionalities. We report the number of times the selected shadow variables were correct within the 20 repetitions of each experiment under different thresholds, as shown in Figure 4.

From the results, we have the following observations and conclusions: (1) In the experiments on the low-dimensional data, changes in $\alpha_1$ have little to no impact on the accuracy of shadow variable selection, while changes in $\alpha_2$ have a significant impact. A decrease in $\alpha_2$ leads to the incorrect exclusion of variables that satisfy Assumption 3.2, resulting in the failure to select correct shadow variables. (2) In the experiments on the high-dimensional data, changes in $\alpha_1$ also have a significant impact on the accuracy of shadow variable selection, similar to $\alpha_2$. It demonstrates that the conditional independence test becomes more challenging as the dimensionality of the covariates increases. (3) As the sample size decreases, the accuracy of shadow variable selection also declines, indicating that both hypothesis tests require a sufficient sample size.

