# OpenReview forum: "Generalizing Causal Effects from Randomized Controlled Trials to Target Populations across Diverse Environments"
_ICML.cc/2025/Conference — ICML 2025 poster_

### Official Review · Reviewer_ogMS · 2025-03-08

**Overall Recommendation:** 3

**Summary:**

This paper studies the identification and estimation problem in generalizing treatment effects from an RCT to an observational dataset. Instead of assuming all relevant separate set is observed, it is assumed that part of them are only observed in one dataset. The identification is made possible by introducing the notion of shadow variables, which is a subset of commonly observed covariates and is conditionally independent of observability given other covariates. The authors then develop the identification theory for the target treatment effect. In addition, they propose a two-stage doubly robust estimator that involves both the selection of shadow variables and doubly robust estimation of the target treatment effect. The efficacy of the proposed methods is demonstrated by extensive numerical experiments.

**Claims And Evidence:**

The experiments are solid, but I have several confusion regarding identification theory and estimation procedures. It is not fully clear to me why the theory goes through under the given assumptions -- please see my questions below.

**Essential References Not Discussed:**

No to my knowledge.

**Experimental Designs Or Analyses:**

Yes, the experiments seem solid.

**Methods And Evaluation Criteria:**

Yes.

**Other Comments Or Suggestions:**

Please see the Questions section below. Given the confusions my current rating is towards reject, but would love to adjust if the authors can address the confusions in the rebuttal.

**Other Strengths And Weaknesses:**

If all results are correct, this paper would be interesting. Strengths include: rich results, new identification conditions for generalization, new procedures. But given the confusions I list in the "questions" section, I would urge the authors to clarify them for me to give a fair judgement.

**Questions For Authors:**

1. Why is Assumption 3.2 testable using observational data, given that $X^m$ is only observed in one dataset?
2. On page 13 (the proof of Lemma 3.4), why Assumptions 2.5 and 3.2 can lead to the first equation (line 664-665)? It seems to use the equivalence between the conditional distributions of $X^m \mid X^c, S=1$ and $X^m \mid X^c$, but I'm not sure this can be implied by the two assumptions, and some clarifications are needed.
3. In Lemma 3.5, what do you mean by "identified"? Do you mean these quantities allow you to compute OR($X^m=x^m, X^c=x^c)$ for every value of $x^m, x^c$? If so, only knowing the conditional expectation $\mathbb{E}[\tilde{OR}(X^m,X^c) \mid X^c,S=1]$ wouldn't allow you to get back to a function of $x^m$ and $x^c$. Please clarify so that I can make a more clear judgement.
4. In step 2 in the identification, why "Under Assumption 2.3, we have $T\indep X^m\mid S=1$"? If I understand correctly, $S=1$ represents the observational dataset in setting 2, so this cannot be true? I think this should be conditional on $S=0$.
5. Can you provide a proof for Theorem 4?
6. Why do you use mean value imputation for $X^m$? Why is this enough for estimating $\tau$ given that Theorem 3.3 uses the density of $X$ given $R=0$?
7. Could you develop consistency and inference guarantees for the resulting estimator?

**Relation To Broader Scientific Literature:**

Generalizing treatment effects to target populations is an important topic, and this paper proposes methods that relax existing conditions, which contributes to this literature and may inspire future developments.

**Theoretical Claims:**

I checked the correctness but cannot arrive at a clear conclusion, and clarification is needed.

---

> ### Author Rebuttal · Authors · 2025-03-31
>
> We sincerely appreciate the reviewer’s great efforts and insightful comments to improve our manuscript. In below, we address these concerns point by point.
>
> > **[Q1]**
>
> Regarding the testability of Assumption 3.2, we would like to clarify the following:
> 1. Assumption 3.2(1), i.e., $Z\notindep X^m |\widetilde{X^c}, S=1$, is testable because **all three variables are observable in the dataset with $S=1$**, where $S=1$ indicates that $X^m$ is observable in this dataset.
> 2. Assumption 3.2(2), i.e., $Z\indep S |X^m, \widetilde{X^c}$, is testable for the following reason. Although $X^m$ is unobservable in the dataset with $S=0$, we can still conduct the test based on Theorem 4.1, which **does not require the values of $X^m$ in the dataset with $S=0$ since the only expression involving $X^m$ in the equation to be solved, i.e., $\frac{S}{Q(X^m, \widetilde{X^c})}$, is always zero when $S=0$**.
>
> > **[Q2]**
>
> Thank you for pointing out the issue caused by our accidental omission of the factor $\frac{P(S=1|X^m,X^c)}{P(S=1|X^c)}$ in the corresponding lines. However, we would like to clarify that this does not affect our final conclusion (the result on lines 674-675). **The corrected proof is as follows:**
>
> $
> E\left[\frac{P(S=0|X^m,X^c)}{P(S=1 |X^m,X^c)}\cdot\frac{P(S=1|X^m=0,X^c)}{P(S=0|X^m=0,X^c)}\middle|X^c,S=1\right]$
> $=\sum_{x^m}\frac{P(S=0|X^m=x^m,X^c)}{P(S=1|X^m=x^m,X^c)} \cdot\frac{P(S=1|X^m=0,X^c)}{P(S=0|X^m=0,X^c)}\cdot P(X^m=x^m|X^c,S=1)$
> $=\sum_{x^m}\frac{P(S=1|X^m=0,X^c)}{P(S=0|X^m=0,X^c)}\cdot\frac{P(S=0|X^m=x^m,X^c)}{P(S=1|X^m=x^m,X^c)}\cdot\frac{P(S=1|X^m=x^m,X^c)}{P(S=1|X^c)}\cdot P(X^m=x^m|X^c)$
> $=\frac{P(S=1|X^m=0,X^c)}{P(S=0|X^m=0,X^c)\cdot P(S=1|X^c)}\cdot\sum_{x^m}P(S=0|X^m=x^m,X^c)\cdot P(X^m=x^m|X^c)\\
> =\frac{P(S=1|X^m=0,X^c)}{P(S=0|X^m=0,X^c)\cdot P(S=1|X^c)}\cdot\sum_{x^m}P(S=0,X^m=x^m|X^c)$
> $=\frac{P(S=1|X^m=0,X^c)\cdot P(S=0|X^c)}{P(S=0|X^m=0,X^c)\cdot P(S=1|X^c)}
> $
>
> Thank you again for your insightful feedback. We will address this issue in the revised version of the manuscript.
>
> > **[Q3]**
>
> **The identification of $OR(X^m,X^c)$ is guaranteed by Theorem 1 in [1], as referenced in Appendix D.** The core idea relies on the completeness condition being satisfied. The detailed proof can be found in the Appendix (Proof of Theorem 1 on Pages 14-15) of [1]. Due to the character limit in the rebuttal stage, we regret that we are unable to provide the full proof here. We kindly ask that the reviewer refer to the original paper for further details. We will include this proof in Appendix D of the revised manuscript. Thank you for the suggestion.
>
> > **[Q4]**
>
> Thank you for pointing out a typo in our manuscript. Indeed, **it should be $T\indep X^m|R=1$**, where $R=1$ is equivalent to $S=0$ in Setting 2. We will correct this typo in the revised version of the manuscript.
>
> > **[Q5]**
>
> As cited in the manuscript, Theorem 4.1 is derived from Theorem 2.3 in [2], and **the detailed proof can be found in Appendix A.3 (Page 12) of [2]**. Due to the character limit in the rebuttal stage, we regret that we are unable to provide the full proof here. We kindly ask that the reviewer refer to the original paper for further details. We will include this proof in the appendix of the revised manuscript. Thank you for your valuable suggestion.
>
> > **[Q6]**
>
> **We employed mean value imputation to reduce the complexity of imputation and improve its practicality and efficiency.** Performing distributional-estimation-based imputation would be perfect, but it requires additional estimation of quantities such as variance, higher-order moments, etc., or the use of generative methods, all of which would increase complexity. Mean value imputation is a commonly used technique to reduce complexity [3], and our experimental results demonstrate that it performs well. Of course, Theorem 3.3 supports distributional-estimation-based imputation, and we will explore it in future work. Thank you for your insightful suggestion.
>
> > **[Q7]**
>
> Due to the character limit in the rebuttal stage, we are unable to include the full proof here. **We promise to provide the consistency proofs for both doubly robust imputation and doubly robust ATE estimation in the appendix of the revised version.** Thank you for your insightful suggestion.
>
> ***
>
> **We hope the above discussion will fully address your concerns about our work, and we would really appreciate it if you could be generous in raising your score. Thank you!**
>
> > **References**
>
> [1] Miao, W., Liu, L., Li, Y., Tchetgen Tchetgen, E. J., & Geng, Z. (2024). Identification and semiparametric efficiency theory of nonignorable missing data with a shadow variable. ACM/JMS Journal of Data Science, 1(2), 1-23.
>
> [2] d’Haultfoeuille, X. (2010). A new instrumental method for dealing with endogenous selection. Journal of Econometrics, 154(1), 1-15.
>
> [3] Lin, W. C., & Tsai, C. F. (2020). Missing value imputation: a review and analysis of the literature (2006–2017). Artificial Intelligence Review, 53, 1487-1509.

---

> > ### Comment · Reviewer_ogMS · 2025-04-04
> >
> > Thank you for the response! I have updated the score.

---

> > > ### Author Response · Authors · 2025-04-07
> > >
> > > We are glad to have addressed your concerns and sincerely appreciate your support for our work, as well as your valuable suggestions for improving it. Thank you!

---

### Official Review · Reviewer_Geaz · 2025-03-14

**Overall Recommendation:** 3

**Summary:**

This paper deals with generalizing treatment effects estimated from RCTs to different environments where there exists environmental shifts. Existing methods assume that covariates common to both source and target datasets contain the separating set, which is often violated in real-world.

The authors propose a Two-Stage Doubly Robust (2SDR) method to address this. The key idea is to relax the standard assumption. Instead of requiring the separating set to be present in the common covariates, they only require it to be present in at least one of the datasets. They then use shadow variables (covariates correlated with the missing ones but not directly influencing the environmental shift) to impute the missing covariates.

The authors also provide theoretical justification and evaluate the method on synthetic and real-world datasets.

**Claims And Evidence:**

1. The theory assumes that the chosen shadow variables are sufficient to fully determine the distribution of the missing covariates. There's no guarantee that such a set of shadow variables exists or that the proposed selection procedure will find them. If the shadow variables are only weakly predictive of the missing covariates, the imputation will be poor, leading to bias.

2. The claim of unbiasedness also depends on the correct specification of the imputation models or the odds ratio function. The paper acknowledges the doubly robust property, but this only protects against misspecification of one model in each pair. It does not guarantee unbiasedness if both models in a pair are misspecified, or if the shadow variable assumption is violated. The empirical evaluation doesn't systematically investigate the impact of model misspecification.

3. The selection procedure depends on the reliability of the conditional independence tests (RCIT). I would recommend authors discuss these tests' own assumption and limitations because these tests are not perfect, especially in high-dimensional settings with limited data.

**Essential References Not Discussed:**

N/A

**Experimental Designs Or Analyses:**

The experiments in general demonstrate the effectiveness of the proposed method.

**Methods And Evaluation Criteria:**

While I know it may be beyond the scope of this rebuttal, simulating scenarios where the shadow variable assumption is partially violated (e.g., by introducing a weak dependence between Z and S) and quantify the resulting bias in the TATE estimates would significantly strengthen the work.

**Other Comments Or Suggestions:**

Figure 1(a) and the related definitions in its caption need improvement. I am confused what type of variables are "covariates affecting treatment effect heterogeneity". Are these variables confounders? Why author state $X_2$ is a covariate affecting neither treatment effect heterogeneity nor environmental shifts in line 68-69?

Furthermore, the wording "variables affect environmental shifts" was misleading. Seems like the covariates exhibit distributional differences, but they don't cause the underlying environmental shift.

**Other Strengths And Weaknesses:**

N/A

**Questions For Authors:**

Refer to all the weaknesses in the "Claims and Evidence" and "Methods and Evaluation Criteria" sections above.

**Relation To Broader Scientific Literature:**

Estimating causal effects from multiple environments is quite important in a lot of the applications, in particular, experiments with multiple sites.

**Theoretical Claims:**

I quickly went over the proof of Theorem 3.3, I did not notice significant flaw in the proof.

---

> ### Author Rebuttal · Authors · 2025-03-31
>
> We sincerely appreciate the reviewer’s great efforts and insightful comments to improve our manuscript. In below, we address these concerns point by point.
>
> > **[Claims And Evidence 1]**
>
> First, we would like to clarify that **the method we propose for automatically selecting shadow variables is theoretically guaranteed, based on the testability of Assumption 3.2**:
> 1. Assumption 3.2(1), i.e., $Z\notindep X^m |\widetilde{X^c}, S=1$, is testable because **all three variables are observable in the dataset with $S=1$**, where $S=1$ indicates that $X^m$ is observable in this dataset.
> 2. Assumption 3.2(2), i.e., $Z\indep S |X^m, \widetilde{X^c}$, is testable for the following reason. Although $X^m$ is unobservable in the dataset with $S=0$, we can still conduct the test based on Theorem 4.1, which **does not require the values of $X^m$ in the dataset with $S=0$ since the only expression involving $X^m$ in the equation to be solved, i.e., $\frac{S}{Q(X^m, \widetilde{X^c})}$, is always zero when $S=0$**.
>
> The shadow variables selected by the proposed method should pass the above hypothesis tests, ensuring that they satisfy Assumption 3.2.
>
> Second, **the existence of shadow variables can also be assessed through hypothesis testing**. Moreover, as discussed in the manuscript, Assumption 3.2 is reasonable because not all variables in $X^c$ with predictive ability for $X^m$ are direct causes of $S$ in many real-world scenarios, such as the example of the AIDS study mentioned in the manuscript. Of course, in certain cases, shadow variables may indeed be weak or even absent. **We have followed the reviewer's suggestion and conducted experiments in such cases**.
>
> > **[Claims And Evidence 2]**
>
> Our method, as most DR methods do, requires that at least one of the two models in both stages, either the regression model or the selection score model, be correctly specified to ensure the consistency of the imputation and ATE estimation. **However, such requirement for correct model specification is very common in related work [1, 2], and we can leverage techniques such as neural networks to learn the correct model specification as accurately as possible.** Of course, we value the reviewer's suggestion and **have also conducted experiments to assess the impact of incorrect model specification due to violations of Assumption 3.2**.
>
> > **[Claims And Evidence 3]**
>
> Thank you for your insightful suggestion. Our method does not rely on specific CIT methods; any CIT method can be used for our hypothesis testing of Assumption 3.2(1). The choice of the method should be determined based on the specific application context. Due to the character limit in the rebuttal phase, we regret that we cannot provide a detailed discussion of the various CIT methods here. **We promise to include the relevant discussion in the appendix of the revised manuscript**.
>
> > **[Methods And Evaluation Criteria]**
>
> Thank you for the suggestion. **We have conducted experiments in cases where Assumption 3.2 is violated.** Specifically, we conducted experiments on the synthetic dataset by
> 1. changing the coefficient of Z on S from 0 to \{0.1, 0.3\} to introduce weak dependence;
> 2. reducing the correlation coefficient between Z and $X^m$ from 0.8 to \{0.3, 0.1\}, so that Z only has a weak predictive ability for $X^m$.
>
> |Extent|Case 1|Case 2|
> |:-:|:-:|:-:|
> |High|0.355$\pm$0.319|0.383$\pm$0.280|
> |Low|0.311$\pm$0.293|0.341$\pm$0.281|
> |None|0.268$\pm$0.212|0.268$\pm$0.212|
>
> The results demonstrates that **while the performance of 2SDR does decline when Assumption 3.2 is violated, the decline is steady as the extent to which Assumption 3.2 is violated increases**.
>
> > **[Other Comments Or Suggestions]**
>
> Thank you for your valuable suggestions.
> 1. **"Covariates affecting treatment effect heterogeneity" are not equivalent to confounders**: Suppose there are three covariates, A, B, and C, and the generation process of Y is $Y=A\cdot T+B+C$. In this case, A is the only variable affecting treatment effect heterogeneity, as the value of $Y(1)-Y(0)$ changes with variations in A. We will improve the related definitions in the figure and its caption.
> 2. **"Variables affect environmental shifts" refers to the variables in the causal graph that point to R (the causes of R)**. We will revise this statement to make it clearer.
>
> ***
> **We hope the above discussion will fully address your concerns about our work, and we would really appreciate it if you could be generous in raising your score. Thank you!**
> > **References**
>
> [1] Miao, W., Liu, L., Li, Y., Tchetgen Tchetgen, E. J., & Geng, Z. (2024). Identification and semiparametric efficiency theory of nonignorable missing data with a shadow variable. ACM/JMS Journal of Data Science, 1(2), 1-23.
>
> [2] Colnet, B., Mayer, I., Chen, G., Dieng, A., Li, R., Varoquaux, G., ... & Yang, S. (2024). Causal inference methods for combining randomized trials and observational studies: a review. Statistical science, 39(1), 165-191.

---

> > ### Comment · Reviewer_Geaz · 2025-04-03
> >
> > Thank you for your rebuttal. My main concerns are addressed. Thus I will raise my score to 3.
> >
> > But as I go through the comments from other reviews to ensure I give a fair judgment on the work, exploring distributional-estimation-based imputation in the paper could enhance robustness.
> >
> > If possible, could you provide the consistency proofs for both doubly robust imputation and doubly robust ATE estimation during the rebuttal period? I understand there exists 5000 character limit, but at least providing sketch proofs here would be really helpful to evaluate the work.

---

> > > ### Author Response · Authors · 2025-04-07
> > >
> > > We are glad that our previous response has addressed your concerns. We greatly appreciate your support of our work, as well as the constructive suggestions you have provided for improving our paper. Below are our responses to your new suggestions:
> > > > **[S1] Exploring distributional-estimation-based imputation**
> > >
> > > Thank you for the valuable suggestion. We use mean value imputation because it is more efficient while remaining effective. We agree that performing distributional-estimation-based imputation could enhance robustness, and our Theorem 3.3 also supports this approach. **We will take your suggestion and explore it in future work.**
> > > > **[S2] Sketch consistency proofs**
> > >
> > > 1. **The DR imputation model $\hat{\psi}$ is consistent if either the $X^m$ regression model or the selection score model of S is correctly specified, and the $\widetilde{OR}$ model is correctly specified.**
> > > **Proof:**
> > > We only need to prove the unbiasedness of $\hat{\psi}$, and then the consistency of $\hat{\psi}$ in large samples holds under the conditions specified in [2,3].
> > > Step 1: Unbiasedness of the OR model.
> > > Given the conditions specified in [1], $\hat{f}(Z|\widetilde{X^c},S)$ is a consistent estimate of $P(Z|\widetilde{X^c},S)$. Therefore, based on Eq.(6), if the $\widetilde{OR}$ model is correctly specified, its unbiasedness holds, and thus the unbiasedness of the OR model also holds based on Eq.(4).
> > > Step 2: Unbiasedness of $\hat{\psi}$.
> > > We need to prove that $E[\hat{\psi}-\psi]=E[(S\cdot(\omega\cdot X^m-\omega\cdot \hat{\delta}(X^c))-\hat{\delta}(X^c))-X^m]=E[(\omega\cdot S-1)\cdot(\hat{\delta}(X^c)-X^m)]=0$ holds, where $\omega=1/P(S=1|\widetilde{X^c},X^m)$. It is equivalent to proving that $E[(\omega\cdot S-1)\cdot(\hat{\delta}(X^c)-X^m)|\widetilde{X^c},X^m]=0$ holds. Under Assumption 3.2(2), this further reduces to proving that $E[\omega\cdot P(S=1|\widetilde{X^c},X^m)-1]\cdot E[\hat{\delta}(X^c)-X^m|\widetilde{X^c},X^m]=0$ holds. We provide proofs for the following two cases.
> > > **(1) The $X^m$ regression model is correctly specified while the selection score model is not.** In this case, as the $X^m$ regression model is correctly specified, $\hat{\theta}$ is an unbiased estimate of $E[X^m|\widetilde{X^c},S=1]$. Therefore, given the unbiasedness of the OR model, based on Eq.(2), the unbiasedness of $\hat{\delta}$ holds. Consequently, $E[\hat{\delta}(X^c)-X^m|\widetilde{X^c},X^m]=0$ holds, and thus the unbiasedness of $\hat{\psi}$ holds.
> > > **(2) The selection score model is correctly specified while the $X^m$ regression is not.** In this case, as the selection score model is correctly specified, $\hat{\gamma}$ is an unbiased estimate of $P(S|\widetilde{X^c})$. Therefore, given the unbiasedness of the OR model, based on Corollary 3.7, the unbiasedness of $\hat{\pi}_s$ holds. Consequently, $E[\omega\cdot P(S=1|\widetilde{X^c},X^m)-1]=0$ holds, and thus the unbiasedness of $\hat{\psi}$ holds.
> > > 2. **If the imputation model is consistent, the DR TATE estimator $\hat{\tau}$ is consistent if either the $Y$ regression model or the selection score model of R is correctly specified, and the $\widetilde{OR}$ model is correctly specified.**
> > > **Proof:**
> > > When either $\hat{\pi}_r$ or $\hat{\mu}_t$ is consistent, the consistency theory of the DR TATE estimator in large samples has already been established in [5]. Therefore, we focus on proving the consistency of $\hat{\pi}_r$ and $\hat{\mu}_t$ here.
> > > (1) If the selection score model and the $\widetilde{OR}$ model are correctly specified, $\hat{\pi}_s$ is consistent, as proved earlier. Since $\hat{\pi}_r$ equals $\hat{\pi}_s$ or $1-\hat{\pi}_s$ based on Corollary 3.7, $\hat{\pi}_r$ is also consistent.
> > > (2) Given that the imputation model is consistent, if the outcome regression model is correctly specified, $\hat{\mu}_t$ estimated using the imputed data still maintains consistency under the conditions specified in [2,5].
> > > Therefore, at least one of $\hat{\pi}_r$ or $\hat{\mu}_t$ is consistent, and thus $\hat{\tau}$ is consistent.
> > >
> > > We will provide full proofs in the revised manuscript.
> > > ***
> > > **We hope the new discussion will fully address your remaining concerns, and we would really appreciate it if you could be generous in raising your score further. Thank you again for your valuable suggestions.**
> > > > **References**
> > >
> > > [1] Silverman, B. W. (2018). Density estimation for statistics and data analysis. Routledge.
> > >
> > > [2] Newey, W. K., & McFadden, D. (1994). Large sample estimation and hypothesis testing. Handbook of econometrics, 4, 2111-2245.
> > >
> > > [3] Miao, W., & Tchetgen Tchetgen, E. J. (2016). On varieties of doubly robust estimators under missingness not at random with a shadow variable. Biometrika, 103(2), 475-482.
> > >
> > > [4] Colnet, B., Josse, J., Varoquaux, G., & Scornet, E. (2022). Causal effect on a target population: a sensitivity analysis to handle missing covariates. Journal of Causal Inference, 10(1), 372-414.
> > >
> > > [5] Little, R. J., & Rubin, D. B. (2019). Statistical analysis with missing data. John Wiley & Sons.

---

### Official Review · Reviewer_fZcZ · 2025-03-27

**Overall Recommendation:** 4

**Summary:**

This paper studies the problem of generalizing RCTs under environment shifts, particularly shifts in the distribution and quantity of covariates. It relaxes the assumption in the prior literature where the separating set (variables that simultaneously affect treatment effect heterogeneity and environmental shifts) is present in both the RCT dataset and target observational dataset. Instead it assumes that the separating set belongs to at least one of the two datasets. Leveraging a shadow variable, the authors introduce a novel Two-Stage Doubly Robust (2SDR) estimator for the Target Average Treatment Effect (TATE). They support their solution through identification theory and demonstrate its effectiveness on both synthetic and real datasets.

**Claims And Evidence:**

(1) The paper extends the TATE identifiability conditions, which traditionally require the separating set to be fully observable in both the RCT and the target observational datasets. Instead, it allows for partial observability, requiring the separating set to be present in at least one of the datasets. This generalization is theoretically supported by a novel identifiability framework.

(2) The authors propose a Two-Stage Doubly Robust (2SDR) TATE estimator that leverages a shadow variable Z. Empirically, the estimator outperforms existing baselines.

**Essential References Not Discussed:**

The relevant literature appears to be sufficiently discussed.

**Experimental Designs Or Analyses:**

I haven’t reviewed the dataset generation procedures in detail, but the overall experimental design and analysis seem appropriate and suggest that 2SDR outperforms the alternatives.

**Methods And Evaluation Criteria:**

Yes, the proposed method appears to be novel, and the evaluation seems to be well-suited and thorough.

**Other Comments Or Suggestions:**

(1) A more detailed practical example for Settings 1 and 2 would be helpful.

(2) Figure 1 is not very clear or readable unless the entire Section 2 is read. There is room for improving the figure by making some of the definitions more explicit.

(3) Readability of Section 4.1.2 needs some improvement (maybe by explaining what is happening in each step or having a diagram).

**Other Strengths And Weaknesses:**

Strengths

(1) The extended TATE framework appears to be novel and well-motivated. The relaxation of the separating set to belong to just one of the datasets offers practical value, as it accounts for potential covariate shifts or missing data when generalizing to other sites.
(2) Experimental results suggest that 2SDR outperforms other baselines on real datasets, particularly in the JTPA dataset experiment, where it improves estimates for sites different from the RCT data.


Weaknesses

(1) Assumption 3.2 seems to be a strong condition that may be difficult to satisfy in practice (see questions below).
(2) A more detailed discussion is needed to convey the intuition behind why the 2SDR estimator performs better.

**Questions For Authors:**

(1) How realistic is it to assume the existence of a shadow variable Z that satisfies Assumption 3.2? Under what conditions can we expect this to hold?

(2) Will results deteriorate smoothly w.r.t. the extent to which this Assumption 3.2 is satisfied?

(3) Typically, under environmental shifts, there may also be shifts in Y|X shifts. It would be beneficial if the authors discussed situations where we expect only X-shifts and where both X- and Y-shifts might occur,  thereby clarifying where 2SDR can be effectively applied?

**Relation To Broader Scientific Literature:**

The paper generalizes the TATE identifiability conditions, which traditionally require the separating set to be fully observable in both the RCT and target observational datasets, to allow for partial observability. In this case, the separating set needs to be a subset of one of the datasets.

In general, generalizing RCT results beyond the sites where they are conducted is an important problem. This paper addresses environmental shifts, particularly changes in the distribution and quantity of covariates, to enhance the generalizability of RCT results.

**Theoretical Claims:**

I have reviewed the proofs of Lemma 3.4 and Lemma 3.5, and they seem to be correct to me.

---

> ### Author Rebuttal · Authors · 2025-03-31
>
> We sincerely appreciate the reviewer’s great efforts and insightful comments to improve our manuscript. In below, we address these concerns point by point.
>
> > **[Other Comments Or Suggestions]**
>
> Thank you for your valuable suggestions. We will carefully revise our manuscript based on your recommendations.
> 1. We will provide more detailed descriptions of the practical examples for Settings 1 and 2.
> 2. We will improve the related definitions in the figure and its caption to make them clearer.
> 3. We will add a brief summary at the beginning of each proof step to explain what is happening in that step, and we will also provide a diagram to further illustrate it.
>
> > **[Q1] How realistic is it to assume the existence of a shadow variable Z that satisfies Assumption 3.2?**
>
> Assumption 3.2 requires that among the common covariates shared by the two datasets, there exist variables that are correlated with the covariates missing in one of the datasets but do not directly influence environmental shifts (not a direct cause of R). **Assumption 3.2 is reasonable in many real-world scenarios, as, typically, most covariates only indirectly affect R, with relatively few variables being direct causes of R.** Moreover, **Assumption 3.2 is testable**:
> 1. Assumption 3.2(1), i.e., $Z\notindep X^m |\widetilde{X^c}, S=1$, is testable because all three variables are observable in the dataset with $S=1$, where $S=1$ indicates that $X^m$ is observable in this dataset.
> 2. Assumption 3.2(2), i.e., $Z\indep S |X^m, \widetilde{X^c}$, is testable for the following reason. Although $X^m$ is unobservable in the dataset with $S=0$, we can still conduct the test based on Theorem 4.1, which does not require the values of $X^m$ in the dataset with $S=0$ since the only expression involving $X^m$ in the equation to be solved, i.e., $\frac{S}{Q(X^m, \widetilde{X^c})}$, is always zero when $S=0$.
>
> Therefore, **we can assess whether Assumption 3.2 holds in real-world applications by conducting hypothesis tests on the covariates**.
>
> > **[Q2] Will results deteriorate smoothly w.r.t. the extent to which this Assumption 3.2 is satisfied?**
>
> Thank you for your insightful suggestion. **We have conducted experiments in cases where Assumption 3.2 is violated.** Specifically, we conducted experiments on the synthetic dataset under the following cases:
> 1. **Case 1: Assumption 3.2(2) is violated.** We changed the coefficient of Z on S from 0 to \{0.1(Low), 0.3(High)\} to introduce weak dependence between Z and S;
> 2. **Case 2: Assumption 3.2(1) is violated.** We reduced the correlation coefficient between Z and $X^m$ from 0.8 to \{0.3(Low), 0.1(High)\}, so that Z only has a weak predictive ability for $X^m$.
>
> |Extent|Case 1|Case 2|
> |:-:|:-:|:-:|
> |High|0.355$\pm$0.319|0.383$\pm$0.280|
> |Low|0.311$\pm$0.293|0.341$\pm$0.281|
> |None|0.268$\pm$0.212|0.268$\pm$0.212|
>
> The results demonstrates that **while the performance of 2SDR does deteriorate when Assumption 3.2 is violated, it deteriorates smoothly as the extent to which Assumption 3.2 is violated increases**. We will include the results and analysis of this experiment in the revised manuscript.
>
> > **[Q3] Typically, under environmental shifts, there may also be shifts in Y|X shifts. It would be beneficial if the authors discussed situations where we expect only X-shifts and where both X- and Y-shifts might occur, thereby clarifying where 2SDR can be effectively applied?**
>
> Thank you for your valuable suggestion. 2SDR relies on Assumption 3.1, which requires that $Y(t)\indep R|X^m,X^c$ holds. Therefore, the applicable situations for 2SDR can be summarized as follows:
> 1. **X-shifts.** As stated in Definition 2.1, the original definition of environmental shift in our work is X-shifts, i.e., $P(X|R=1)\neq P(X|R=0)$.
> 2. **Both X- and Y-shifts caused by X.** Although we did not explicitly state this in Definition 2.1, due to the fact that variables in X may be causes of Y, the distribution of Y will also shift along with the distribution of X, i.e., $P(Y|R=1)\neq P(Y|R=0)$, or alternatively, $P(X,Y|R=1)\neq P(X,Y|R=0)$.
> 3. **$Y|X^c$ shifts.** Given that $Y(t)\indep R|X$ holds, we have $P(Y|X,R=1)=P(Y|X,R=0)$. Therefore, our problem setting essentially assumes the absence of Y|X shifts. However, for the common covariates $X^c$ shared by the two datasets, there still exist $Y|X^c$ shifts, i.e., $P(Y|X^c,R=1)\neq P(Y|X^c,R=0)$. Assumption 3.2 required by 2SDR is still satisfied under $Y|X^c$ shifts, and therefore, 2SDR can address $Y|X^c$ shifts. However, existing methods rely on Assumption 2.8, which requires that $Y(t)\indep R|X^c$ holds. Under $Y|X^c$ shifts, Assumption 2.8 does not hold, and thus, **in contrast to 2SDR, previous methods cannot address $Y|X^c$ shifts**.
>
> We will include the above discussion in the revised manuscript.
>
> ***
> **We hope the above discussion will fully address your concerns about our work, and we would really appreciate it if you could be generous in raising your score. Thank you!**

---

> > ### Comment · Reviewer_fZcZ · 2025-04-03
> >
> > Thanks for your detailed response. My concerns have been appropriately addressed, I have raised my score to 4.

---

> > > ### Author Response · Authors · 2025-04-07
> > >
> > > We are glad to have addressed your concerns and sincerely appreciate your support for our work, as well as your valuable suggestions for improving it. Thank you!

---

### Decision · Program_Chairs · 2025-05-01

**Decision:**

Accept (poster)

**Comment:**

This paper introduces a two-stage method for generalizing treatment effects from RCTs to target populations facing environmental shifts. Unlike common solutions, the proposed procedure relaxes the assumption that the separating set of covariates is fully observable in both the RCT and target data. Instead, it only requires the separating set to be present in at least one of the two data groups and utilizes shadow variables to impute missing covariates in a two-stage process.

Reviewers generally acknowledged the novelty and importance of the problem and found the experimental results promising, particularly on real-world datasets like JTPA. A key concern was the realism regarding the existence and properties of shadow variables, which was partially addressed by the author response. More detailed and practical examples will significantly strengthen the paper further.